# MOLM: MIXTURE OF LoRA MARKERS

**Samar Fares[1], Nurbek Tastan[1], Noor Hussein[2], Karthik Nandakumar[1,2]**
[1]Mohamed bin Zayed University of Artificial Intelligence (MBZUAI), UAE
[2]Michigan State University (MSU), USA
`samar.fares@mbzuai.ac.ae, nandakum@msu.edu`

## ABSTRACT

Generative models can generate photorealistic images at scale. This raises serious concerns about the ability to detect synthetically generated images and attribute these images to specific sources. While watermarking has emerged as a possible solution, existing methods remain fragile to realistic distortions, susceptible to adaptive removal, and expensive to update when the underlying watermarking key changes. We propose a general watermarking framework that formulates the encoding problem as key-dependent perturbation of the parameters of a generative model. Within this framework, we introduce Mixture of LoRA Markers (MOLM), a routing-based instantiation in which binary keys activate lightweight low-rank adapters (LoRA) inside residual and attention blocks. This design avoids key-specific re-training and achieves the desired properties such as imperceptibility, fidelity, verifiability, and robustness. Experiments on Stable Diffusion and FLUX show that MOLM preserves image quality while achieving robust key recovery against distortions, compression and regeneration, averaging attacks, and black-box adversarial attacks on the extractor. Code is available at https://github.com/Samar-Fares/MOLM-Watermark.

## 1 INTRODUCTION

Recent advances in diffusion models have enabled unprecedented progress in text-to-image generation, with models such as Stable Diffusion (Rombach et al., 2022) and FLUX (Labs, 2024) producing high-quality, photorealistic outputs at scale. While these models provide powerful creative and practical tools, their ability to synthesize realistic content raises concerns regarding authenticity, misuse, and attribution. To address these challenges, watermarking has emerged as a core strategy for enabling model owners to verify whether a given image originated from their model. An effective watermark must satisfy four criteria: *imperceptibility* (the watermark does not degrade image quality), *fidelity preservation* (outputs remain close to the real image distribution), *robustness* (the watermark resists removal and forgery), and *detection/attribution* (the watermark can be reliably extracted and linked to the model of origin).

Existing watermarking methods face persistent challenges. **First**, the WAVES benchmark (An et al., 2024) shows that while some watermarks survive minor distortions, adversarial attacks easily break them. Regeneration attacks (Zhao et al., 2024b) erase watermarks by denoising and reconstructing images, affecting methods like Tree-Ring (Wen et al., 2023) and Stable Signature (Fernandez et al., 2023). Averaging attacks (Yang et al., 2024a) remove or forge content-agnostic watermarks by combining generated samples, which has been demonstrated on Tree-Ring (Wen et al., 2023), Gaussian Shading (Yang et al., 2024b), and Stable Signature (Fernandez et al., 2023). Surrogate decoders (Jiang et al., 2023) craft perturbations that bypass the target extractor while preserving recovery in a shadow model, and purification-based defenses (Saberi et al., 2024) can erase watermarks with minimal perceptual change. **Second**, robustness often conflicts with perceptual quality: improving resilience typically introduces visible degradation (Zhao et al., 2024a), and modifying the initial noise reduces image quality. **Third**, many approaches are computationally expensive and inflexible. Methods embedding watermarks in the latent space or training process demand costly retraining, especially when updating or changing watermarking keys. Examples include full-model finetuning in backdoor methods (Liu et al., 2023; Zhao et al., 2023), weight modulation in WOUAF (Kim et al., 2024), per-key training in Stable Signature (Fernandez et al., 2023) and SleeperMark (Souri

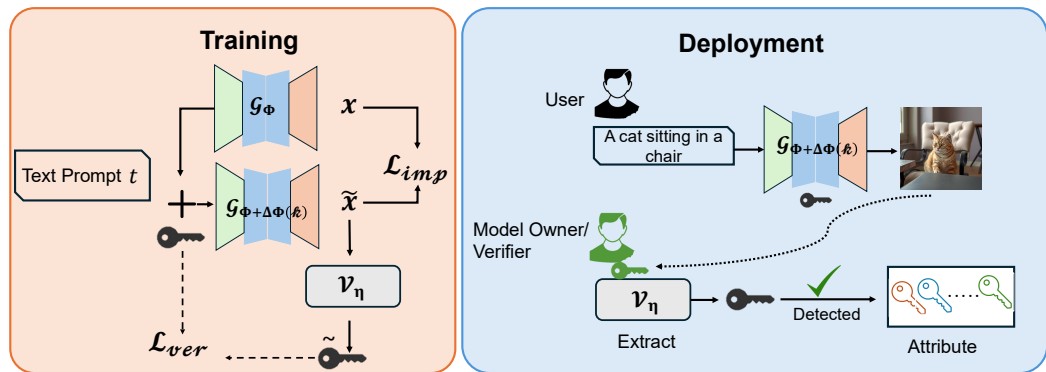

Figure 1: **Proposed watermarking framework.** During training (left), a text prompt **t** is processed by both the frozen generator $\mathcal{G}_\Phi$ (producing clean image **x**) and the perturbed generator $\mathcal{G}_{\Phi+\Delta\Phi(\kappa)}$ (producing watermarked image $\tilde{\mathbf{x}}$), where $\Delta\Phi(\kappa)$ denotes a key-dependent parameter perturbation. The extractor $\mathcal{V}_\eta$ recovers the embedded key $\tilde{\kappa}$. The parameter perturbation function ($\Delta\Phi$) and extractor parameters ($\eta$) are jointly learned by optimizing the perceptual loss $\mathcal{L}_{\text{imp}}$ and key loss $\mathcal{L}_{\text{ver}}$. During deployment (right), the model owner/verifier extracts the embedded key to verify the watermark and attribute generated images to a specific model.

et al., 2022), per-prompt optimization in ROBIN (Huang et al., 2025), and pretraining in AquaLoRA (Feng et al., 2024). Such overhead limits their practicality in dynamic deployments.

In this paper, we make two main contributions: First, we introduce a general watermarking framework that formulates watermarking as key-dependent perturbations of a frozen generative model. Second, building on this framework, we propose Mixture of LoRA Markers (MOLM), a routing-based instantiation that reinterprets low-rank adapters (LoRA) as watermark carriers. A binary key deterministically selects adapter activations across generative model building blocks, embedding watermark information without modifying the backbone. MOLM offers three advantages: (i) Efficiency - only lightweight adapters are trained; no pretraining on watermarked data is required, and the model itself produces watermarked samples during generation; (ii) Scalability - capacity scales naturally with the number of routing layers and adapter choices, without retraining for new keys; and (iii) Robustness - distributed routing makes the watermark harder to remove or forge while preserving image fidelity. We evaluate MOLM on Stable Diffusion and FLUX across the MS-COCO, LAION-Aesthetics datasets, testing fidelity, extraction accuracy, capacity scaling, and robustness against a wide range of distortions and adversarial attacks. Our results show that MOLM achieves strong key recovery $> 0.98$ bit accuracy, preserves image quality (FID degradation $\leq 1.5$), and maintains robustness under geometric and photometric distortions, compression and diffusion-based removal, averaging attacks, and adaptive adversarial attacks on the extractor. These findings highlight MOLM as a practical and scalable watermarking method for modern diffusion models.

## 2 BACKGROUND

### 2.1 DIFFUSION MODELS

Diffusion models are a leading class of generative models, delivering state-of-the-art image quality with stable training (Ho et al., 2020). They rely on two processes: a forward pass that gradually adds Gaussian noise to data, and a reverse pass that learns to denoise and recover samples. Latent Diffusion Models (LDMs) (Rombach et al., 2022) improve efficiency by operating in a compressed latent space rather than pixel space, with an encoder–decoder pair mapping between images and latents ($z = E(x)$, $\hat{x} = D(z)$). A U-Net backbone (Ronneberger et al., 2015) acts as the denoiser, extracting multi-scale features during the reverse process. Guidance mechanisms such as Classifier-Free Guidance (CFG) (Ho & Salimans, 2021) further enhance controllability. Additional details are provided in Appendix A.1.

## 2.2 Watermarking of Generative Models

Watermarking can be categorized into *model-specific watermarking*, which embeds signals directly via generative models, and *data-specific watermarking*, which modifies input data (Wang et al., 2024a). Model-specific watermarking methods fall into three main types: **Encoder-decoder methods**: These introduce an injector (encoder) to embed messages and a decoder to recover them. Hidden (Zhu et al., 2018) first proposed embedding watermarks as adversarial perturbations, where an encoder perturbs images to encode a secret and a decoder extracts it without the original. For GANs, (Yu et al., 2021) embedded artificial fingerprints into training data, while (Fei et al., 2022) trained GANs with a fixed decoder and auxiliary watermark loss to enforce ownership. (Zeng et al., 2023) replaced the decoder with a detector, adversarially training the injector–detector pair. **Backdoor-based methods**: Inspired by backdoor attacks, these embed a trigger that activates watermark generation. (Zhao et al., 2023) bound a token to a watermark image via DreamBooth fine-tuning (Ruiz et al., 2023). (Liu et al., 2023) proposed naiveWM, linking a trigger word directly to a watermark, and fixedWM, requiring a fixed prompt position for stealth. SleeperMark (Wang et al., 2024b) injects latent-level watermarks via UNet fine-tuning with triggered prompts. **Generation-process methods**: These modify the diffusion process itself, embedding watermarks into latent trajectories or model components. Tree-Ring (Wen et al., 2023) inserts concentric patterns in Fourier latents, while ROBIN (Huang et al., 2025) jointly optimizes patterns with text conditioning. Gaussian Shading (Yang et al., 2024b) maps watermark bits into latents without training, with extraction via DDIM inversion. (Xiong et al., 2023) fused binary matrices into decoder layers, while Rezaei et al. (2024) and Meng et al. (2024) progressively embedded across latent layers. Peng et al. (2023) proposed WDP, where a parallel diffusion trajectory yields verifiable watermarked samples. Stable Signature (Fernandez et al., 2023) fine-tuned the LDM decoder with an extractor to enforce multi-bit signatures. WOUAF (Kim et al., 2024) modulated decoder weights via affine transformations of watermark messages with a joint decoder. AquaLoRA (Feng et al., 2024) embedded watermarks into UNet LoRA modules by pretraining a latent watermark with encoder–decoder and fine-tuning with a prior-preserving loss.

## 3 Proposed Method

**Notations**: Let $\mathcal{G}_\Phi : \mathcal{Q} \times \mathcal{T} \to \mathcal{X}$ be a text-to-image generation model that starts with a random initialization (latent) $\mathbf{q} \in \mathcal{Q}$ and uses the conditioning text (prompt) $\mathbf{t} \in \mathcal{T}$ to generate an image $\mathbf{x} \in \mathcal{X}$. Here, $\mathcal{G}$ denotes the architecture of the generative model, $\Phi$ denotes its parameters, and $\mathcal{Q}, \mathcal{T}$, and $\mathcal{X}$ represent the latent, prompt, and image spaces, respectively. A watermarking system $\mathcal{W}$ consists of a tuple $(\mathcal{U}_\zeta, \mathcal{V}_\eta)$ that represents the watermark embedding and extraction algorithms, respectively. The watermark embedding method $\mathcal{U}_\zeta : \mathcal{X} \times \mathcal{K} \to \mathcal{X}$ can be considered as a wrapper around the generator $\mathcal{G}_\Phi$ that embeds a given $M$-bit key $\kappa \in \mathcal{K} \subseteq \{0,1\}^M$ within the generated image to create a watermarked image $\tilde{\mathbf{x}}$. The watermark extractor $\mathcal{V}_\eta : \mathcal{X} \to \mathcal{K}$ is designed to extract a key $\tilde{\kappa} \in \mathcal{K}$ from a given image. Let $\mathbf{x} = \mathcal{G}_\Phi(\mathbf{q}, \mathbf{t})$ be an image generated by the model $\mathcal{G}_\Phi$ for some $\mathbf{q} \in \mathcal{Q}$ and $\mathbf{t} \in \mathcal{T}$. Let $\tilde{\mathbf{x}} = \mathcal{U}_\zeta(\mathbf{x}, \kappa)$ be the watermarked image for some $\kappa \in \mathcal{K}$. Let $\tilde{\kappa} = \mathcal{V}_\eta(\tilde{\mathbf{x}})$ be the extracted key. Ideally, the watermarking system must satisfy the following four properties: (i) *Imperceptibility*: The watermarked image $\tilde{\mathbf{x}}$ is as close as possible to the original generated image $\mathbf{x}$; (ii) *Fidelity*: The distribution of $\tilde{\mathbf{x}}$ is as close as possible to the distribution of real images; (iii) *Verifiability*: The extracted key $\tilde{\kappa}$ is as close as possible to the embedded key $\kappa$; and (iv) *Robustness*: It should not be possible to remove the key $\kappa$ from $\tilde{\mathbf{x}}$ without significantly degrading its visual content (removal attack) or add the key $\kappa$ to an image (could be real or synthetic) not generated using $\mathcal{G}_\Phi$) (forgery attack).

Note that the above watermarking system $\mathcal{W}$ can be used either for *detection* or *attribution*. A watermark detector certifies that a given image contains a valid watermark if the extracted key $\tilde{\kappa}$ is sufficiently close to a known key $\kappa$, i.e., if $d(\kappa, \tilde{\kappa}) \leq \epsilon$, where $d$ is a distance (e.g., Hamming) metric and $\epsilon$ is the detection threshold. On the other hand, an attribution method stores a database of models or users along with their corresponding keys. The attributor searches for the closest match between the extracted key and the stored keys in the database to determine which specific model or user generated the watermarked image.

**Problem Setting**: In this work, we consider the following setting with four players: (a) **Model Owner** owns the generative model $\mathcal{G}_\Phi$ (e.g., LDM) and the watermarking system $\mathcal{W} = (\mathcal{U}_\zeta, \mathcal{V}_\eta)$.

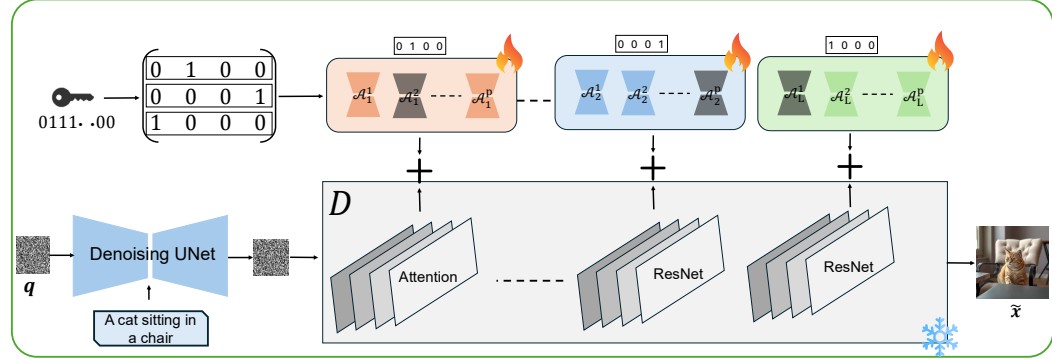

Figure 2: **MOLM generation pipeline.** A binary key $\kappa$ is mapped into a routing collection $\{s_\ell\}_{\ell \in [L]}$ that determines the active LoRA adapters $\{\mathcal{A}_\ell^{(s_\ell)}\}$ across ResNet and Attention blocks of the frozen generator. During the diffusion sampling process, this routing implements the perturbation $\Delta\Phi(\kappa)$, yielding the watermarked image $\tilde{\mathbf{x}} = \mathcal{G}_{\Phi + \Delta\Phi(\kappa)}(\mathbf{t})$ for a given prompt $\mathbf{t}$. The backbone weights $\Phi$ remain frozen, ensuring negligible added inference cost.

When presented with a text prompt $\mathbf{t} \in \mathcal{T}$ from the user, the model owner randomly samples $\mathbf{q} \in \mathcal{Q}$ and $\kappa \in \mathcal{K}$ to generate a watermarked image $\tilde{\mathbf{x}} = \mathcal{U}_\zeta(\mathcal{G}_\Phi(\mathbf{q}, \mathbf{t}), \kappa)$ and outputs this watermarked image to the user. (b) **User**: The user interacts with the generative model through an API, providing prompts $\mathbf{t}$ and receiving watermarked images as outputs $\tilde{\mathbf{x}}$. The user has no knowledge about the generative model or the watermarking system. (c) **Verifier**: The watermark verifier has access only to the watermark extractor $\mathcal{V}_\eta$. Given an image $\mathbf{x} \in \mathcal{X}$ and a key $\kappa$ (or a set of keys in attribution mode), the verifier detects if the given image contains a valid watermark (or attributes the image to a specific model or user). While we envision the verifier to be a trusted third-party (with whom the model owner shares the required watermarking keys), it is also possible for the model owner to double up as the verifier. (d) **Adversary**: The goal of the adversary is to circumvent the watermarking system through *removal* and *forgery* attacks. While the adversary may have high-level knowledge about the nature of the generative model and/or watermarking system, the adversary does not have access to any of the model parameters ($\Phi, \zeta, \eta$). In the removal attack, the adversary attempts to modify a valid watermarked image $\tilde{\mathbf{x}}$ so that the verifier detects it as a non-watermarked image. These modifications could either be simple image transformations (e.g., cropping, rotation, brightness adjustment, or JPEG compression) or obtained through black-box adversarial attacks on the watermark detector. In the forgery attack, the adversary attempts to modify a non-watermarked image (could be real or synthetically generated using some other generative model) so that the verifier detects it as a watermarked image. To aid such an attack, the adversary may collect a set of valid watermarked images by querying the generative model via the API.

**Problem Statement**: Given a generative model $\mathcal{G}_\Phi$, our goal is to design a watermarking system $\mathcal{W}$ (see Figure 1) that strongly satisfies the four required properties (imperceptibility, fidelity, verifiability, and robustness). Furthermore, we want to dynamically embed high-entropy keys (measured by the key size $M$) into the watermarked images without requiring any key-specific re-training. Specifically, our watermark encoder $\mathcal{U}_\zeta$ is modeled as a perturbation of parameters of the generative model, i.e., $\mathcal{U}_\zeta(\mathcal{G}_\Phi(\mathbf{q}, \mathbf{t}), \kappa) = \mathcal{G}_{\Phi + \Delta\Phi(\kappa)}(\mathbf{q}, \mathbf{t})$. Given a set of training samples $\{(\mathbf{t}_n, \mathbf{x}_n)\}_{n=1}^N$, where $\mathbf{x}_n = \mathcal{G}_\Phi(\mathbf{q}, \mathbf{t}_n)$ (here, $\mathbf{q}$ is randomly sampled), the task is to learn a suitable mapping from the key to the parameter perturbation space along with the corresponding watermark extractor $\mathcal{V}_\eta$.

## 3.1 MOLM: MIXTURE OF LORA MARKERS

**Low-Rank Adaptation (LoRA)** (Hu et al., 2022) is a parameter-efficient fine-tuning technique originally proposed to adapt foundation models for specific downstream tasks. Instead of updating the full set of model parameters, LoRA performs low-rank decomposition of parameter changes. Hence, it freezes the original weight matrices and injects trainable low-rank matrices parallel to the existing layers. More details about LoRA are available in Appendix A.2. In our work, we

do not use LoRA for traditional model adaptation, but reinterpret it as a mechanism to achieve key-dependent perturbation $\Delta\Phi(\kappa)$ of the generative model parameters. Furthermore, our work is inspired by the idea of learning a Mixture of LoRA Experts (MoLE) (Wu et al., 2024) to adapt large models for diverse tasks. In MoLE, multiple experts/adapters are added in parallel to the layers of the given model and the input is dynamically routed through a subset of these experts based on a data-dependent gating mechanism. While MoLE focuses on data-driven expert composition, we leverage this idea to implement key-dependent expert composition.

**Watermark Encoder.** The given generative model $\mathcal{G}_\Phi$ can be considered as a sequence of $\bar{L}$ blocks $\mathcal{F}_1, \mathcal{F}_2, \cdots, \mathcal{F}_{\bar{L}}$, i.e., $\mathcal{G}_\Phi(.) = (\mathcal{F}_{\bar{L}} \circ \mathcal{F}_{\bar{L}-1} \circ \cdots \mathcal{F}_1)(.)$, where $\circ$ denotes function composition. Let $\mathbf{h}_{\ell-1}$ denote the input to block $\mathcal{F}_\ell$ and $\mathbf{h}_\ell$ denote its output. We deterministically preselect a subset of $L$ blocks from the set $\{\mathcal{F}_1, \mathcal{F}_2, \cdots, \mathcal{F}_{\bar{L}}\}$ and add $P$ low-rank adapters $\{\mathcal{A}_\ell^{(1)}, \mathcal{A}_\ell^{(2)}, \cdots, \mathcal{A}_\ell^{(P)}\}$ to each selected block $\ell \in [L]$. Thus, $(L \times P)$ low-rank adapters are added to the model architecture and learned. However, during image generation, we activate only one adapter $\mathcal{A}_\ell^{(s_\ell)}$ for each chosen block, where $s_\ell \in \{1, \ldots, P\}$ is the selected adapter index. Thus, the operations involved in a selected block $\ell$ can be expressed as:

$$\mathbf{h}_\ell = \mathcal{F}_\ell(\mathbf{h}_{\ell-1}) + \alpha\, \mathcal{A}_\ell^{(s_\ell)}(\mathbf{h}_{\ell-1}), \tag{1}$$

where $\alpha$ is a fixed scaling factor. Note that for the unselected blocks $\mathbf{h}_\ell = \mathcal{F}_\ell(\mathbf{h}_{\ell-1})$. Let $\psi_\ell^p$ denote the parameters of the adapter $\mathcal{A}_\ell^{(p)}$ (the $p^{\text{th}}$ adapter in the $\ell^{\text{th}}$ block) and $\Psi = \{\psi_\ell^p\}$, where $p \in [P]$ and $\ell \in [L]$, denote the set of all additional parameters added to the generative model.

The critical aspect of the proposed method is how the adapters are dynamically selected during image generation based on the given watermark key $\kappa$. Firstly, the $M$-bit binary key $\kappa$ is broken down into $L$ non-overlapping chunks $\kappa_1, \kappa_2, \cdots, \kappa_L$, where each chunk $\kappa_\ell$ contains $\log_2 P$ bits. In our implementation, the value of $P$ is always limited to a power of 2 and $M$ is set to $(L \cdot \log_2 P)$ bits. The chunk $\kappa_\ell$ is assigned to block $\ell$, where $\ell \in [L]$, and is converted into the corresponding decimal index $s_\ell \in [P]$. The collection $\{s_\ell\}_{\ell \in [L]}$ defines the key-specific routing path through the mixture of low-rank adapters. Since this routing path directly determines the watermarking signal embedded in the generated image, we refer to the proposed framework as a mixture of LoRA markers (MOLM). Thus, watermark encoding is achieved by augmenting the parameters $\Phi$ of the generator with a set of LoRA markers ($\Psi(\kappa)$) selected based on the watermark key $\kappa$, i.e., $\Delta\Phi(\kappa) = \Psi(\kappa) \subset \Psi$.

**Watermark Extractor.** The watermark extractor $\mathcal{V}_\eta$ is a deep neural network that takes an image $\mathbf{x}$ as input and produces $M$ logits $\boldsymbol{u} = [u_1, \ldots, u_M]$, which are passed through a sigmoid function $\sigma$ to yield a continuous approximation of the extracted key $\hat{\kappa} = \sigma(\boldsymbol{u})$. Note that the binary extracted key $\tilde{\kappa}$ can be easily obtained by rounding the values in $\hat{\kappa}$ to either 0 or 1.

**Training.** Recall that the two main requirements of the watermarking system are imperceptibility (which also indirectly ensures fidelity if we assume that the original generative model already has high fidelity) and verifiability. Therefore, we employ two loss functions to enforce these constraints. First, we apply the perceptual loss $\mathcal{L}_{\text{imp}}$, instantiated as a feature-based reconstruction loss between the watermarked image $\tilde{\mathbf{x}}_n = \mathcal{G}_{\Phi+\Psi(\kappa)}(\mathbf{q}, \mathbf{t}_n)$ and its corresponding non-watermarked image $\mathbf{x}_n = \mathcal{G}_\Phi(\mathbf{q}, \mathbf{t}_n)$ generated by the same model using the same prompt $\mathbf{t}_n$ and latent instantiation $\mathbf{q}$.

$$\mathcal{L}_{\text{imp}} = \mathbb{E}_{\kappa \sim \mathcal{K}} \frac{1}{N} \sum_{n=1}^{N} \sum_{k=1}^{K} w_k \left\| \varphi_k(\mathcal{G}_{\Phi+\Psi(\kappa)}(\mathbf{q}, \mathbf{t}_n)) - \varphi_k(\mathcal{G}_\Phi(\mathbf{q}, \mathbf{t}_n)) \right\|_2^2, \tag{2}$$

where $\{\varphi_k\}_{k \in [K]}$ are fixed perceptual feature extractors (e.g., LPIPS) and $w_k$ are the relative weights assigned to them.

The watermark extractor parameters $\eta$ are trained using the binary cross-entropy loss: $\mathcal{L}_{\text{ver}}$:

$$\mathcal{L}_{\text{ver}} = \mathbb{E}_{T \sim \Pi} \frac{1}{N} \sum_{n=1}^{N} \left[ \frac{1}{M} \sum_{m=1}^{M} \left( -\kappa_m \log \sigma(u_m) - (1 - \kappa_m) \log(1 - \sigma(u_m)) \right) \right], \tag{3}$$

where $T \sim \Pi$ denotes image-space augmentations of the watermarked image for robustness, $\boldsymbol{u} = \mathcal{V}_\eta(T(\tilde{\mathbf{x}}_n))$, and $\kappa_m$ and $u_m$ are the components of $\kappa$ and $\boldsymbol{u}$, respectively.

Thus, the overall training objective combines these two losses as follows:

$$\min_{\Psi,\eta} \quad \left[ \mathcal{L}_{\text{ver}} + \lambda \, \mathcal{L}_{\text{imp}} \right], \tag{4}$$

where $\Psi$ are the parameters of the LoRA markers, $\eta$ are the watermark extractor parameters, and $\lambda \geq 0$ balances between the two losses.

**Deployment.** As illustrated in Figure 2, a key $\kappa$ deterministically selects a routing path $\{s_\ell\}_{\ell \in [L]}$ through the mixture of LoRA markers, which in turn defines $\Psi(\kappa) \subset \Psi$. For a prompt $\mathbf{t}$ and random latent initialization $\mathbf{q}$, the perturbed generator then produces the watermarked image as:

$$\tilde{\mathbf{x}} = \mathcal{G}_{\Phi + \Delta\Phi(\kappa)}(\mathbf{q}, \mathbf{t}), \tag{5}$$

where $\Delta\Phi(\kappa) = \Psi(\kappa)$ is realized via the activated LoRA markers. The routing mask remains fixed across the denoising trajectory, ensuring that the same key always induces the same execution path and hence, produces an extractable watermark. Importantly, MOLM does not alter the backbone sampling procedure and introduces negligible cost at inference time. At the time of verification, the given image is passed through the watermark extractor $\mathcal{V}_\eta$ and the extracted key is validated.

## 4 EXPERIMENTS

### 4.1 IMPLEMENTATION DETAILS

Modern text-to-image generative models are typically implemented as diffusion models with two key components: a U-Net denoising network and a decoder network (often based on a variational autoencoder (VAE)). Both are composed of modular ResNet and Attention layers, which we refer to as *routing layers*. Each ResNet block itself contains multiple convolutional sub-layers, while attention blocks implement self- or cross-attention. In our main implementation, each ResNet block is treated as a single routing layer.

### 4.2 EXPERIMENTAL SETUP

We train on 10k image–text pairs (Zhai et al., 2023) sampled from the MS-COCO 2014 dataset (Lin et al., 2014). For text-to-image generation evaluation, we use the PNDM scheduler (Liu et al., 2022) with $T = 50$ denoising steps. The CFG scale is set to 7.5 unless otherwise specified. For evaluation, we generate images from the MS-COCO test set prompts as well as captions randomly sampled from the LAION-Aesthetics dataset (Schuhmann et al., 2022). We evaluate MOLM on two diffusion models: (i) Stable Diffusion (SD) v1.5 (Rombach et al., 2022), generating $512 \times 512$ images, and (ii) FLUX (Labs, 2024), a recent large-scale diffusion model generating $1024 \times 1024$ images. We additionally report qualitative results on SD v3.5. By default, we perturb the parameters of ResNet blocks in the VAE decoder with key-dependent LoRA adapters, i.e., each block is treated as a single routing layer. Unless otherwise stated, we activate 14 such residual routing layers with $P = 4$ adapters per layer (corresponding to 2 bits/layer), resulting in 28-bit keys. We also experimented with perturbing the U-Net parameters, which contains 22 ResNet blocks, 16 cross-attention layers, and 16 self-attention layers. This configuration yields a total of 108 bits. However, as discussed in the Appendix D.1, this led to noticeable degradation in generation quality. We leave perturbation of U-Net parameters for future exploration, as larger key sizes must be balanced with fidelity constraints. More details on the experimental setup can be found in the Appendix B.1.

### 4.3 EVALUATION METRICS

For fidelity, we compute PSNR and SSIM (Wang et al., 2004) between images generated with and without watermarking. To assess distributional quality, we report the Fréchet Inception Distance (FID) (Heusel et al., 2017) between generated samples and real images from the MS-COCO validation set. For key recovery, we report the average bit accuracy, defined as the proportion of correctly decoded key bits across watermarked images:

$$\text{Bit Accuracy} = \frac{s(\kappa, \tilde{\kappa})}{M}, \text{where } s(\kappa, \tilde{\kappa}) = \sum_{j=1}^{M} \mathbb{1}[\kappa_j = \tilde{\kappa}_j]. \tag{6}$$

For watermark detection, similar to (Fernandez et al., 2023), we perform a hypothesis test based on the number of matching bits $s(\kappa, \tilde{\kappa})$. The input is declared a valid watermarked image if $s(\kappa, \tilde{\kappa}) \geq \tau$. Under $H_0$ (no watermark), $s \sim \text{Binomial}(M, 0.5)$, and $\text{FPR}(\tau) = P(s \geq \tau \mid H_0)$. We set $\tau$ to achieve a target FPR (e.g., 1%), and report TPR@1%FPR on watermarked images. For our default $M = 28$ setting, this corresponds to $\tau = 20$ matching bits.

For the attribution task, we consider a database of users, each with an $M$-bit key $\kappa^{(i)} \in \{0,1\}^M$. We match the key $\hat{\kappa}$ extracted from an image with each registered key by computing the number of matching bits, $s_i = \sum_{j=1}^{M} \mathbb{1}\{\hat{\kappa}_j = \kappa_j^{(i)}\}$. An image is attributed to user $\hat{i} = \arg\max_i s_i$ if $\max_i s_i \geq \tau$. Otherwise, it is rejected as non-watermarked. We evaluate attribution performance using three complementary metrics: (i) False positive rate (the proportion of non-watermarked images that are not rejected), (ii) True accept rate (the proportion of legitimate watermarked images that are not rejected) and (iii) Conditional attribution accuracy (the fraction of watermarked images that correctly matched to their originating user, conditional on the fact that they are not rejected).

Table 1: **Detection and robustness results.** We report fidelity on undistorted watermarked images (FID ↓, SSIM ↑, PSNR ↑) and detection robustness under common distortions. **Bit Accuracy** ↑ and **TPR@1% FPR** ↑ are used as the evaluation metrics for the bit-recovery (**Top**) and detection-only methods (**Bottom**), respectively. Values highlighted in Orange denote the difference with respect to the corresponding non-watermarked images. Within each column, **bold** indicates the best and the second-best value. Robustness results are averaged over two distortion severity levels (see App. E).

| Model | Data | Method | FID ↓ | SSIM ↑ | PSNR ↑ | \multicolumn{6}{c}{Robustness (Detection accuracy under distortions) ↑} | Key Size |
|---|---|---|---|---|---|---|---|---|---|---|---|---|
| | | | | | | Undistorted | Crop | Rot | Res | Bright | JPEG | |
| \multicolumn{13}{l}{*Bit-Recovery Methods (Bit Accuracy)*} |
| SD | MS-COCO | Stable Signature | 29.5 (+0.4) | **0.85** | **27.8** | **0.99** | **0.97** | 0.56 | 0.72 | **0.95** | 0.89 | 48 |
| | | AquaLoRA | 30.5 (+1.4) | 0.63 | 22.1 | 0.95 | 0.91 | 0.45 | **0.91** | 0.72 | **0.94** | 48 |
| | | WOUAF | **27.8** (−1.3) | 0.73 | 24.9 | 0.98 | **0.96** | **0.85** | 0.71 | **0.98** | **0.98** | 32 |
| | | MOLM (Ours) | **27.7** (−1.4) | 0.77 | 23.5 | 0.98 | 0.91 | **0.84** | 0.90 | 0.95 | 0.89 | 28 |
| | LAION | Stable Signature | 74.9 (+5.5) | **0.80** | 26.0 | 0.98 | **0.97** | 0.57 | 0.72 | **0.96** | 0.89 | 48 |
| | | WOUAF | **69.8** (+0.4) | 0.70 | 24.3 | 0.98 | 0.91 | **0.65** | 0.65 | **0.94** | 0.64 | 32 |
| | | MOLM (Ours) | **69.5** (+0.1) | 0.65 | 22.3 | 0.93 | 0.90 | **0.84** | **0.87** | 0.92 | **0.90** | 28 |
| FLUX | MS-COCO | Stable Signature | 26.2 (−0.8) | 0.85 | 23.4 | **0.98** | **0.98** | 0.56 | 0.67 | **0.97** | 0.81 | 48 |
| | | MOLM (Ours) | **25.8** (−1.2) | **0.95** | **32.3** | 0.93 | 0.89 | **0.76** | **0.82** | 0.92 | 0.77 | 28 |
| SD 3.5 | MS-COCO | MOLM (Ours) | 26.2 (−0.04) | 0.97 | 36.9 | 0.91 | 0.81 | 0.66 | 0.79 | 0.85 | 0.69 | 28 |
| \multicolumn{13}{l}{*Detection-Only Methods (TPR@1%FPR)*} |
| SD | MS-COCO | Tree-Ring | 30.9 (+1.8) | 0.45 | 13.0 | **0.99** | **0.99** | **0.98** | **0.99** | **0.99** | **0.99** | − |
| | | Gaussian-Shading | **24.3** (−4.8) | 0.21 | 8.7 | **1.00** | 0.67 | 0.48 | **1.00** | **1.00** | **1.00** | − |
| | | ROBIN | **25.3** (−3.8) | 0.73 | 22.1 | **1.00** | **0.99** | 0.64 | − | 0.93 | **0.99** | − |
| | | MOLM (Ours) | 27.7 (−1.4) | 0.77 | 23.5 | **1.00** | 0.98 | **0.98** | 0.96 | **0.99** | **0.99** | − |
| | LAION | Tree-Ring | 76.1 (+6.7) | **0.42** | 12.6 | 0.97 | **0.97** | **0.97** | **0.97** | 0.96 | **0.99** | − |
| | | Gaussian-Shading | **66.4** (−2.9) | 0.19 | 8.6 | **0.99** | 0.50 | 0.0 | **0.99** | **0.99** | **0.99** | − |
| | | MOLM (Ours) | **69.5** (+0.1) | 0.65 | 22.3 | **0.99** | 0.95 | **0.98** | 0.94 | **0.99** | **0.99** | − |
| FLUX | MS-COCO | MOLM (Ours) | 25.8 (−1.2) | 0.95 | 32.3 | 0.98 | 0.98 | 0.92 | 0.96 | 0.97 | 0.90 | − |
| \multicolumn{13}{l}{*Post-hoc Watermarking Methods*} |
| - | - | TrustMark_Q*(TPR;(FPR))* | 28.61 (−0.5) | 0.98 | 40.9 | 0.99(2.38%) | 0.00(2.53%) | 0.00(2.64%) | 0.99(2.25%) | 0.53(2.08%) | 0.99(2.31%) | 100 |
| - | - | VINE-R*(Bit Accuracy)* | 29.86 (+0.8) | 0.99 | 36.5 | 1.00 | 0.52 | 0.51 | 1.00 | 0.95 | 1.00 | 100 |

## 4.4 DETECTION AND ROBUSTNESS RESULTS

Table 1 presents a comparison of MOLM with representative watermarking baselines on Stable Diffusion v1.5 (SD) and FLUX. We group prior methods into two categories: **Bit-Recovery methods**, which embed explicit binary keys and are evaluated based on bit accuracy, namely Stable Signature (Fernandez et al., 2023), AquaLoRA (Feng et al., 2024), and WOUAF (Kim et al., 2024); and **Detection-Only methods**, which provide binary watermark presence/absence decisions and are evaluated based on TPR@1%FPR, namely Tree-Ring (Wen et al., 2023), ROBIN (Huang et al., 2025), and Gaussian-Shading (Yang et al., 2024b). MOLM achieves strong detection performance with bit accuracy above $0.98$ on undistorted images and robust key recovery across all tested distortions. Details about the distortions considered are in Appendix E. All results in Table 1 are averaged over two severity levels for each distortion. MOLM retains $0.91$ accuracy under cropping and $0.89$ under JPEG compression. In contrast, Stable Signature suffers significant drops under rotation ($0.56$) and resizing ($0.72$), while AquaLoRA exhibits poor rotation robustness ($0.45$). WOUAF performs competitively under certain distortions but at a higher training cost (Table 4). These results highlight the advantage of MOLM's routing-based embedding in maintaining consistent key recovery across perturbations. MOLM also compares favorably to detection-only baselines. Tree-Ring and Gaussian-Shading achieve strong TPR for undistorted or mild distorted images. However, both require full inversion during verification. ROBIN introduces adversarial optimization but struggles under rotation ($0.64$). By contrast, MOLM achieves a high TPR ($\geq 0.95$) while simulta-

Table 2: **Attribution results**. **True accept rate** is the fraction of watermarked images that are not rejected. **Conditional attribution accuracy** is the fraction of watermarked images that are correctly attributed to the true user, conditional on the fact that they are not rejected.

| Distortion | True accept rate (%) ↑ | Conditional attribution accuracy (%) ↑ |
|---|---|---|
| None (undistorted) | 98.92 | 100.00 |
| JPEG ($q = 80$) | 96.87 | 100.00 |
| JPEG ($q = 50$) | 88.76 | 100.00 |
| Crop (5%) | 89.09 | 99.99 |
| Crop (10%) | 39.92 | 99.97 |
| Resize (0.7×) | 93.68 | 100.00 |
| Resize (0.3×) | 0.96 | 97.92 |
| Rotation (25°) | 90.53 | 100.00 |
| Rotation (90°) | 0.03 | 66.67 |
| Brightness (×1.5) | 85.16 | 100.00 |
| Brightness (×2.0) | 51.79 | 99.98 |

neously enabling explicit key recovery. On FLUX, MOLM maintains consistent detection accuracy (TPR $\geq 0.96$), confirming that it generalizes beyond Stable Diffusion. Similarly, testing on LAION Aesthetics with models trained on MS-COCO shows that MOLM's watermark remains recoverable even under distribution shifts. This suggests that MOLM is not tied to a single dataset or architecture, but instead provides a transferable mechanism for embedding and detecting watermarks in generative models. We also report results for post-hoc watermarking methods designed for arbitrary images, TrustMark (Bui et al., 2023) and VINE (Lu et al., 2024). Overall, while these post-hoc methods can be robust to mild photometric distortions, they remain vulnerable to geometric attacks. In Appendix B.2 we highlight computational efficiency. MOLM trains within ~1 day on a single A100 with no per-key retraining. At inference time, MOLM introduces negligible overhead beyond the frozen generator, ensuring practical deployability.

## 4.5 ATTRIBUTION RESULTS

We simulate an attribution scenario with 1,000 registered users. For each user, we generate 10 watermarked images using their assigned $M = 28$-bit key, yielding a test set of 10,000 in-database images. We set the acceptance threshold $\tau = 27$ (i.e., at most one bit error permitted) to target a global false positive rate of $10^{-3}$. Under this operating point, MOLM achieves a 0.02% false positive rate when tested on 20,000 non-watermarked images. MOLM also achieves 98.92% true accept rate on undistorted watermarked images and a conditional attribution accuracy of 100%, indicating that the primary source of error is rejection rather than incorrect attribution. This demonstrates that MOLM reliably distinguishes between watermarked and non-watermarked images while maintaining near-perfect user attribution when a watermark is detected. We evaluate robustness by applying the same distortion suite used in Section 4.4. Table 2 reports the true accept rate and conditional attribution accuracy for each distortion type. Common image edits preserve attribution integrity: JPEG compression (quality 80), mild cropping (5%), resizing (0.7×), moderate rotation (25°), and brightness adjustment (×1.5, ×2.0) all achieve acceptance rates above 85% with near-perfect accuracy among accepted images. Severe transformations, extreme resizing (0.3×) and 90° rotation are rejected, with acceptance rates below 1%. Our choice of $\tau = 27$ (one-bit tolerance) is intentionally conservative, prioritizing low false positive rates for high-stakes applications. In deployments where false negatives are more costly, the threshold can be relaxed to $\tau = 26$ or lower, increasing acceptance rates at the cost of a higher false positive rate.

## 4.6 IMAGE GENERATION QUALITY

We assess the impact of watermarking on the perceptual quality of generated images. Quantitatively, we achieve FID values comparable to or lower than existing watermarking methods (Table 1), indicating minimal degradation. Importantly, the differences relative to vanilla Stable Diffusion are

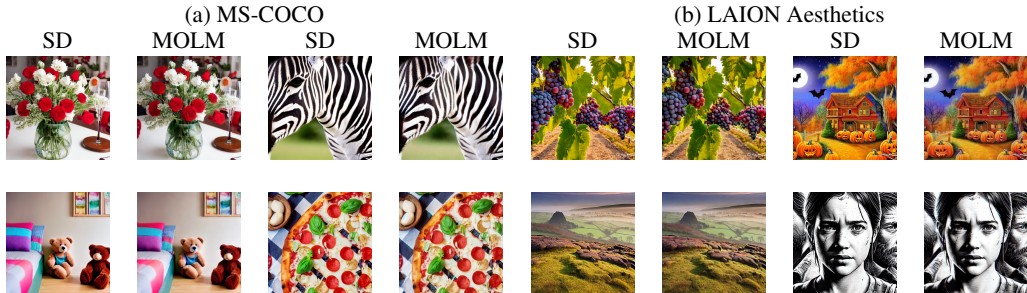

Figure 3: **Image generation quality.** Visual comparison between Stable Diffusion (SD) and MOLM on MS-COCO (left four columns) and LAION Aesthetics (right four columns). For each prompt, we show the original Stable Diffusion image and the corresponding watermarked image. MOLM preserves high image quality.

Table 3: **Robustness under compression, diffusion, and adversarial removal attacks.** We report Bit Accuracy ($\uparrow$) and FID ($\downarrow$). $q$ is the compression quality (lower = stronger), "steps" denotes denoising iterations, and $\varepsilon$ is the MSE bound for PGD attacks.

| Attack | Params | No-Aug | | Aug-Trained | |
|---|---|---|---|---|---|
| | | Bit Acc. | FID | Bit Acc. | FID |
| BMSHJ2018 | $q = 1, 4, 8$ | 0.50/0.77/0.96 | 29.9/28.2/27.8 | 0.61/0.95/0.99 | 30.6/28.6/28.4 |
| Cheng2020 | $q = 1, 3, 6$ | 0.70/0.82/0.93 | 29.2/27.7/27.6 | 0.94/0.95/0.97 | 30.1/28.9/28.7 |
| Diffusion Regen. | steps $= 30, 60, 100$ | 0.72/0.69/0.62 | 27.8/28.4/28.5 | 0.85/0.85/0.82 | 30.2/29.9/31.2 |
| Adversarial (PGD) | $\varepsilon = 10^{-3}, 10^{-2}, 10^{-1}$ | 0.93/0.81/0.60 | 27.7/27.8/28.4 | 1.00/0.99/0.96 | 28.4/28.6/29.0 |

small ($\leq 1.5$ FID), with no systematic drop in SSIM or PSNR. Qualitative comparisons are shown in Figure 3. On both MS-COCO and LAION Aesthetics, MOLM outputs remain visually indistinguishable from baseline Stable Diffusion generations. The routing mechanism does not introduce noticeable artifacts. Additional visual examples are provided in the Appendix E.

## 4.7 WATERMARK ROBUSTNESS

### 4.7.1 COMPRESSION AND DIFFUSION-BASED REMOVAL ATTACKS

Recent work (Zhao et al., 2024b) shows that invisible image watermarks can be removed via compression and diffusion-based regeneration attacks. We evaluate three families of attacks: two learned compression models (bmshj2018 (Ballé et al., 2018) and cheng2020 (Cheng et al., 2020)) and a diffusion/noise attacker, and report both bit accuracy and FID of the attacked images in Table 3. Results are shown for an extractor trained *without* augmentation ("no-aug") and *with* augmentation ("aug-trained"). Augmentation training enhances robustness to compression and diffusion removal attacks.

### 4.7.2 AVERAGING ATTACKS

Yang et al. (Yang et al., 2024a) recently proposed an effective attack against several modern watermarks. Their key observation is that many watermarking schemes across multiple generations maintain a consistent watermark while the image content varies. By averaging $k$ generated images, an adversary can estimate and subtract the watermark for removal, or induce watermark signals into clean images for forgery. We follow their evaluation protocol by generating sets of $k \in \{5, 10, 20, 50, 100, 200, 500, 1000, 2000, 5000\}$ images and computing averages in both grey-box and black-box settings. We then test two attack variants: Removal and Forgery. We also consider a "same-message" setting where all averaged images embed the identical key. Figures 4, and 5 (Appendix C.1) compare MOLM with WOUAF under averaging attacks in both the same-message and heterogeneous-message cases. Under forgery attempts, MOLM remains at chance accuracy ($\approx 0.5$) even at higher $k$ values, even in the same-message case. Under removal attempts, MOLM maintains high bit accuracy ($\geq 0.96$) even when averaging up to 5000 images.

### 4.7.3 ADVERSARIAL ATTACKS ON KEY EXTRACTION (WHITE-BOX)

To evaluate robustness against adaptive adversaries, we evaluate white-box attacks that directly optimize the input image against the watermark extractor. The adversary has full access to the extractor parameters and detection threshold $\tau$, and may perturb an image subject to a perceptual constraint. In our experiments, this constraint is expressed as an upper bound on the mean squared error (MSE) with respect to a reference image; equivalently, it can be stated as a lower bound on PSNR. Details on this attack can be found in the Appendix C.2. As shown in Table 3, adversarial PGD attacks under an MSE constraint $\varepsilon$ significantly reduce key accuracy when no augmentation is used during training (dropping from 0.93 to 0.60 as $\varepsilon$ increases). With augmentation, however, MOLM remains robust, achieving $> 0.96$ accuracy even for $\varepsilon$ values corresponding to PSNR $\approx 10$. Constraints looser than this threshold lead to visible distortions, indicating that successful removal requires perceptual degradation. We also explore a full-knowledge adversary who retrains the generative model independently; the results in Appendix C.3 show that such attacker-generated images yield a bit accuracy of $\approx 0.5$ when using our extractor, i.e., indistinguishable from random guessing.

### 4.8 ABLATION STUDIES

**Key Capacity.** The capacity of MOLM is determined by the number of blocks $L$ selected from the backbone $\mathcal{G}_\Phi$ and the number of available adapter paths $P$ per block. Since choosing among $P$ adapters requires $\log_2 P$ bits, the total key size is $M = L \cdot \log_2 P$. In our default configuration, we perturb the whole $L = 14$ ResNet blocks in the VAE decoder with $P = 4$ adapters each, yielding $M = 14 \times 2 = 28$ bits per image. Each ResNet block is treated as a single routing layer, where one index $s_\ell$ determines the active convolutional adapters within the block. We also investigate several extensions: (i) Independent bits per convolution, which doubles capacity by assigning separate bits to the two convolutional layers inside each ResNet block; (ii) Including attention blocks, where routing cross- and self-attention layers increases $L$ and thus the total key size; (iii) Expanding adapter paths, e.g., increasing $P$ from 4 to 8 so that each block encodes $\log_2 8 = 3$ bits. Appendix D.2 reports the empirical trade-offs between capacity, bit accuracy, and fidelity under these configurations.

**Mapping Effect.** To analyze how key bits are distributed across routing adapters, we conduct a layer-wise weight randomization (details in the Appendix D.3). Averaging results over 100 random prompts reveals that the mapping is largely distributed: many routing layers influence multiple bits with intermediate probability, and individual bits are affected by several layers. While certain bits are more sensitive, there is no strict one-to-one correspondence between adapters and bits. Instead, the key is redundantly encoded across layers, which enhances robustness, as perturbing any single adapter does not deterministically erase the watermark.

**Sampling Configurations.** We evaluate watermark robustness under variations in generation settings, including scheduler type, number of denoising steps, and CFG scale. Results in Appendix D.4 show consistently high bit accuracy (0.94–0.96) across all configurations, with only modest variation in FID (25.8–28.5). This demonstrates that MOLM does not depend on a fixed sampler for detection.

**Adapter Rank.** Finally, we investigate the effect of LoRA adapter rank on watermark fidelity and bit accuracy. Table in Appendix D.4.1 shows that higher ranks yield stronger bit recovery and fidelity, while very low ranks (8) fail to sustain reliable decoding. This highlights a trade-off between efficiency and watermark robustness.

## 5 CONCLUSION

We presented a general watermarking framework that views watermark embedding as a key-dependent parameter perturbation of a frozen generative model, and instantiated it with MOLM, a routing-based design that uses lightweight LoRA adapters as watermark carriers. By encoding keys through deterministic adapter selection across decoder blocks. Across Stable Diffusion and FLUX, MOLM preserves generation quality and achieves reliable key extraction, remaining robust to common image edits, compression, diffusion-based regeneration, averaging attacks, and adaptive attacks on the extractor. Together, these results support MOLM as a practical watermarking mechanism for modern text-to-image systems.

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

# Contents

# A    TECHNIQUES

## A.1    DIFFUSION MODELS

The framework of diffusion models is based on two complementary processes: a *forward process* that progressively adds Gaussian noise to a data sample, and a *reverse process* that learns to recover the original sample by removing noise. This process can be formulated as a Markov chain, where the forward process is defined as:

$$q(x_t|x_{t-1}) = \mathcal{N}(x_t; \sqrt{1 - \beta_t}\, x_{t-1}, \beta_t I), \tag{7}$$

where $\beta_t$ represents a noise schedule controlling the amount of noise added at each step (Sohl-Dickstein et al., 2015). The reverse process is parameterized by a neural network trained via variational inference, approximating:

$$p_\theta(x_{t-1}|x_t) = \mathcal{N}(x_{t-1}; \mu_\theta(x_t, t), \Sigma_\theta(x_t, t)). \tag{8}$$

The model is trained to minimize the variational lower bound on the negative log-likelihood, optimizing a series of KL-divergence terms to ensure stable sample reconstruction (Ho et al., 2020).

While diffusion models offer high sample quality, their main limitation lies in slow sampling, as generating an image requires iterating through many timesteps. To accelerate sampling, Denoising Diffusion Implicit Models (DDIMs) introduce a non-Markovian reformulation of the reverse process, enabling deterministic sampling with fewer steps, given by:

$$x_{t-1} = \sqrt{\alpha_{t-1}}x_0 + \sqrt{1 - \alpha_{t-1}}\,\epsilon_t. \tag{9}$$

Here, $\alpha_t$ controls the noise schedule, allowing the model to balance between speed and sample quality (Song et al., 2021).

Recent advancements have further improved diffusion models by introducing guidance mechanisms to control sample generation. Classifier Guidance (Dhariwal & Nichol, 2021) incorporates an external classifier to steer the generation process, while Classifier-Free Guidance (Ho & Salimans, 2021) removes dependency on external classifiers by training the model to generate both conditional and unconditional outputs, allowing for controlled sampling.

To improve efficiency, Latent Diffusion Models (LDMs) (Rombach et al., 2022) reduce computational cost by applying diffusion in a compressed latent space rather than pixel space. LDMs leverage an encoder-decoder architecture, where data is first mapped to a lower-dimensional representation:

$$z = E(x), \quad \tilde{x} = D(z), \tag{10}$$

and the diffusion process operates on the latent $z$ instead of the full-resolution image. This approach significantly reduces memory and computational demands while preserving high-quality outputs. The U-Net architectures (Ronneberger et al., 2015) play a crucial role in diffusion models by providing multi-scale feature representations, balancing fine-grained detail retention with computational efficiency.

## A.2    LOW-RANK ADAPTATION (LORA)

For a linear or convolutional layer with weight matrix $W \in \mathbb{R}^{d_{\text{out}} \times d_{\text{in}}}$, LoRA introduces a residual term $W_{\text{LoRA}} = BA$, where $A \in \mathbb{R}^{r \times d_{\text{in}}}$ and $B \in \mathbb{R}^{d_{\text{out}} \times r}$ are trainable parameters with $r \ll \min(d_{\text{in}}, d_{\text{out}})$. The modified output becomes:

$$W\boldsymbol{x} \quad \longrightarrow \quad W\boldsymbol{x} + \alpha \cdot BA\boldsymbol{x}, \tag{11}$$

where $\alpha$ is a scaling factor. This design enables fine-tuning with significantly fewer parameters while preserving the pre-trained model's behavior.

# B    SETUP

## B.1    EXPERIMENTAL SETUP

The LoRA adapters and key extractor are trained for 12K steps using the AdamW optimizer with a learning rate of $1 \times 10^{-4}$, batch size 4, and weight decay 0.01. $\mathcal{L}_{\text{ver}}$ is implemented as binary

cross-entropy between the extractor's predicted logits and the ground-truth key bits, while $\mathcal{L}_{\text{imp}}$ is an LPIPS loss (VGG backbone) computed between the generated watermarked image and the generated non-watermarked image. The loss is balanced with $\lambda = 1.0$.

## B.2 EFFICIENCY

Table 4 provides detailed training and inference costs across baselines.

Table 4: **Training and inference cost across methods.** We report one-time pre-training cost, and per-key training cost.

| Method | Pre-train Cost | Per-key Cost |
|---|---|---|
| Stable Signature | $\sim 8$ day | $\sim 1$ min/key |
| AquaLoRA | $\sim 2.5$ days | None |
| WOUAF | $\sim 3.5$ days | None |
| Tree-Ring | None | N/A |
| ROBIN | per prompt | N/A |
| Gaussian-Shading | None | N/A |
| MOLM (Ours) | $\sim 1$ day | None |

## C ROBUSTNESS

### C.1 AVERAGING ATTACKS

Figure 4 provides the comparison between MOLM and WOUAF under averaging attacks in the same-message setting. Figure 5 provides MOLM bit accuracy under averaging attacks in the heterogeneous-message setting.

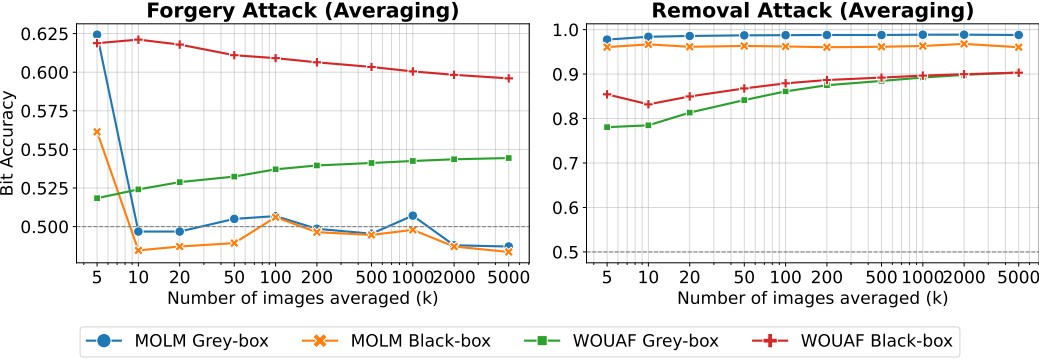

Figure 4: Averaging attack evaluation: MOLM vs. WOUAF (same message). (Left) Forgery attack. MOLM stays at the chance level ($\sim 0.5$). (Right) Removal attack. MOLM achieves accuracy $\geq 0.96$, whereas WOUAF degrades to $\sim 0.85$–$0.90$.

### C.2 ADVERSARIAL ATTACKS ON KEY EXTRACTION (WHITE-BOX)

We consider an adaptive white-box adversary that perturbs a watermarked image to induce failure of the verifier while remaining within a perceptual budget. Given a watermarked image $\tilde{\mathbf{x}} \in [0,1]^{3 \times H \times W}$ and an MSE budget $\varepsilon$, the adversarial example $\mathbf{x}_{\text{adv}}$ is constrained to

$$\text{MSE}(\mathbf{x}_{\text{adv}}, \tilde{\mathbf{x}}) \leq \varepsilon, \qquad \text{MSE}(\mathbf{x}_{\text{adv}}, \tilde{\mathbf{x}}) = \frac{1}{N} \|\mathbf{x}_{\text{adv}} - \tilde{\mathbf{x}}\|_2^2, \tag{12}$$

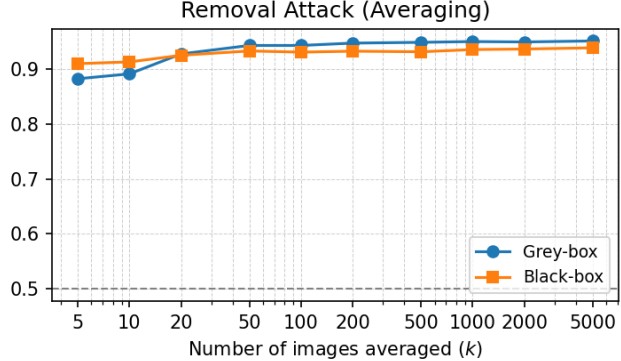

Figure 5: Averaging removal attack evaluation: MOLM (heterogeneous message).

where $N = 3HW$. The adversary aims to make $V_\eta$ uninformative (i.e., outputs close to random guessing). We optimize the removal loss

$$\mathcal{L}_{\mathrm{rem}}(\mathbf{x}_{\mathrm{adv}}) = \frac{1}{M} \sum_{i=1}^{M} \left( p_i(\mathbf{x}_{\mathrm{adv}}) - \tfrac{1}{2} \right)^2. \tag{13}$$

We solve the problem using $\ell_2$-projected gradient descent (PGD). An attack is deemed successful if the verifier's similarity score falls below the detection threshold.

### C.3 FULL KNOWLEDGE ATTACK SCENARIO

We consider a white-box adversary who knows the entire MOLM pipeline (architecture, losses, and training hyperparameters) and is allowed to retrain the generative model on a different subset of the same training data, using different random seeds. The attacker trains their own version and then evaluates the attacker-generated images using our watermark extractor. Across our experiments, our extractor recovers bits from attacker images with average bit-accuracy $\approx 0.5$, i.e., no better than random guessing. We attribute this failure to the fact that the adapters and the extractor are trained end-to-end, the extractor learns to decode the specific routing-induced activation patterns produced by that training run. Small changes in seed, data order, or subset yield different routing statistics, so an independently trained attacker produces images whose routing signature the defender's extractor cannot decode.

## D ABLATIONS

### D.1 PERTURBING THE U-NET

In addition to perturbing the VAE decoder, we experimented with perturbing the U-Net. The U-Net in Stable Diffusion v1.5 consists of 22 ResNet blocks, 16 cross-attention layers, and 16 self-attention layers. Treating each of these modules as routing layers, with $P = 4$ adapters per layer (2 bits each), results in a total of 108 key bits per image. The image quality degraded significantly. FID increased by $4.15$ points, and human inspection revealed a change in the content compared to non-watermarked images as shown in Figure 6, especially around fine textures and edges. This suggests that the heavier intervention in the U-Net perturbs denoising dynamics more strongly than in the decoder, amplifying the perceptual cost. These results indicate that while perturbing the U-Net offers higher key capacity, it compromises fidelity. This trade-off highlights the importance of carefully balancing watermark capacity against image quality.

### D.2 KEY CAPACITY

Table 5 reports the quantitative trade-offs between key size, bit accuracy, detection performance (TPR@1%FPR), and image quality (FID) under different architectural configurations. Table 6 shows visual examples.

| Stable Diffusion | Decoder MOLM (28-bit) | U-Net MOLM (108-bit) |
|:---:|:---:|:---:|
| 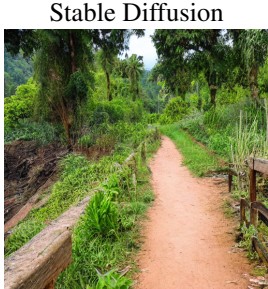 | 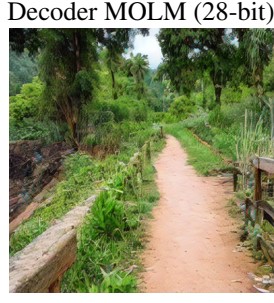 | 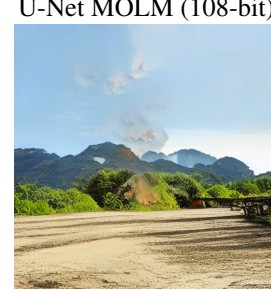 |

Figure 6: **Perturbing the U-Net.** Comparison between Stable Diffusion (left), MOLM with decoder parameters perturbation (28 bits, middle), and MOLM with U-Net perturbation (108 bits, right). While capacity increases, perturbing the U-Net introduces visible artifacts and degrades fidelity.

Table 5: **Key capacity scaling in MOLM.** We report the effective key size (bits), Bit Accuracy ($\uparrow$), TPR@1%FPR ($\uparrow$), and FID ($\downarrow$) for different architectural configurations.

| Configuration | Key Size | Bit Accuracy | TPR@1%FPR | FID |
|---|---|---|---|---|
| Independent bits per convolution | 56 | 0.89 | 0.99 | 27.9 |
| Including attention blocks | 30 | 0.92 | 0.99 | 28.2 |
| 8 paths per layer (3 bits/block) | 42 | 0.90 | 0.99 | 27.3 |

### D.3 MAPPING EFFECT

Starting from a watermarked image $\tilde{\mathbf{x}}$, we iterate over routing blocks. For each block $b$, we identify the active adapter path from the routing mask, temporarily replace its weights with random values of the same shape, regenerate the image using the unchanged routing mask, and run the trained extractor to obtain a predicted key. The original weights are then restored before moving to the next block. Let $\kappa \in \{0,1\}^M$ denote the embedded key and $\tilde{\kappa}^{(b)}$ the extractor output after perturbing block $b$. We define the binary flip matrix

$$F_{b,j} = \mathbb{1}\{\tilde{k}_j^{(b)} \neq k_j\}, \tag{14}$$

where $F_{b,j} = 1$ indicates that bit $j$ flipped when block $b$ was randomized. Figure 7 shows the flip heatmaps

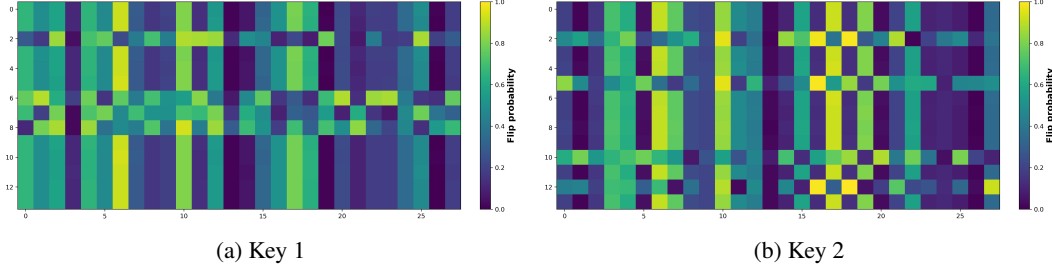

|  (a) Key 1  |  (b) Key 2  |

Figure 7: Bit-flip heatmaps for two different embedded keys, each averaged over 100 random prompts. Rows correspond to routing blocks and columns to key bits; brighter cells indicate a higher probability that the bit flips when the corresponding block is perturbed.

### D.4 SAMPLING CONFIGURATIONS

Table 7 reports the results. Across all tested configurations, MOLM consistently achieves high bit accuracy (0.94–0.96), demonstrating robustness to changes in the sampling procedure.

Table 6: **Qualitative examples for key capacity configurations in MOLM.** Rows are different sampled outputs for the same MS-COCO prompt; columns are configurations.

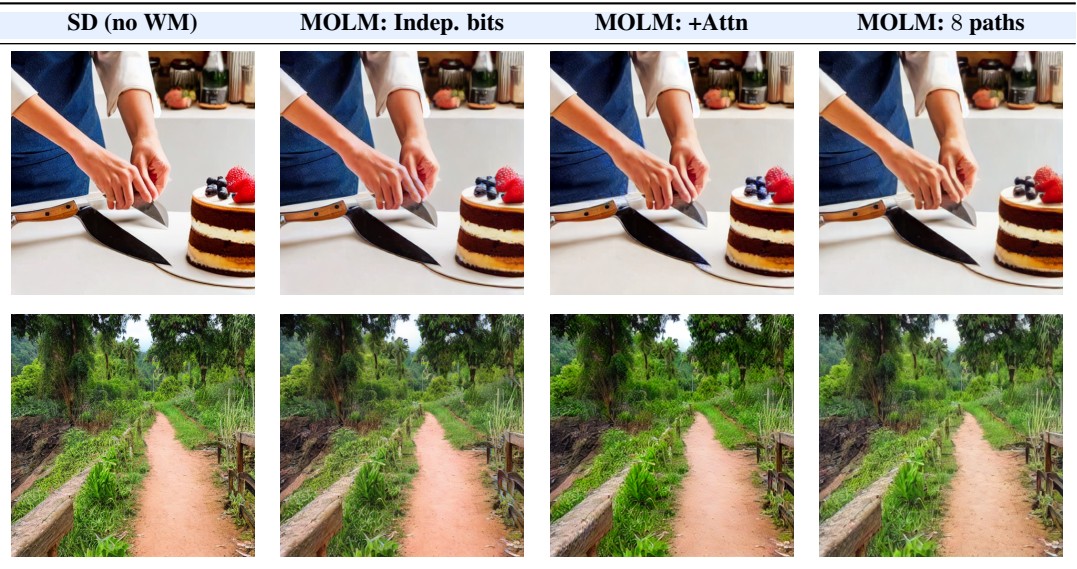

| SD (no WM) | MOLM: Indep. bits | MOLM: +Attn | MOLM: 8 paths |
|---|---|---|---|

### D.4.1 ADAPTER RANK

Table 7 reports the FID of watermarked images and the bit accuracy of the trained extractor for ranks $\{64, 32, 16, 8\}$. Higher ranks yield a stronger watermark recovery (higher bit accuracy), while very low ranks (8) cannot sustain reliable bit decoding.

## E    MORE VISUAL EXAMPLES

Tables 8, 9, 10, 11, 12, and 13 provide extended visual examples covering different models, distortion scenarios, comparisons with existing methods, and ablations on sampling configurations and LoRA rank.

Table 7: **Sampling and ablation studies. Left:** Effect of sampling configurations on MOLM. Bit accuracy remains consistently high ($\geq 0.94$), while FID varies modestly across settings. **Right:** Ablation on the rank of LoRA adapters. Higher ranks improve bit recovery but slightly increase FID.

| Factor | Setting | Bit Accuracy ↑ | FID ↓ |
|---|---|---|---|
| Sampler | DDIM | 0.96 | 27.4 |
| | DPM-S | 0.96 | 28.5 |
| | DPM-M | 0.95 | 27.3 |
| | Euler | 0.94 | 26.5 |
| Steps | 15 | 0.95 | 27.3 |
| | 25 | 0.95 | 25.8 |
| | 100 | 0.95 | 27.4 |
| CFG | 5.0 | 0.96 | 25.9 |
| | 10.0 | 0.95 | 27.6 |

| Rank | FID ↓ | Bit Accuracy ↑ |
|---|---|---|
| 64 | 27.7 | 0.98 |
| 32 | 28.2 | 0.96 |
| 16 | 29.5 | 0.91 |
| 8 | 34.8 | 0.75 |

FLUX                                    MOLM

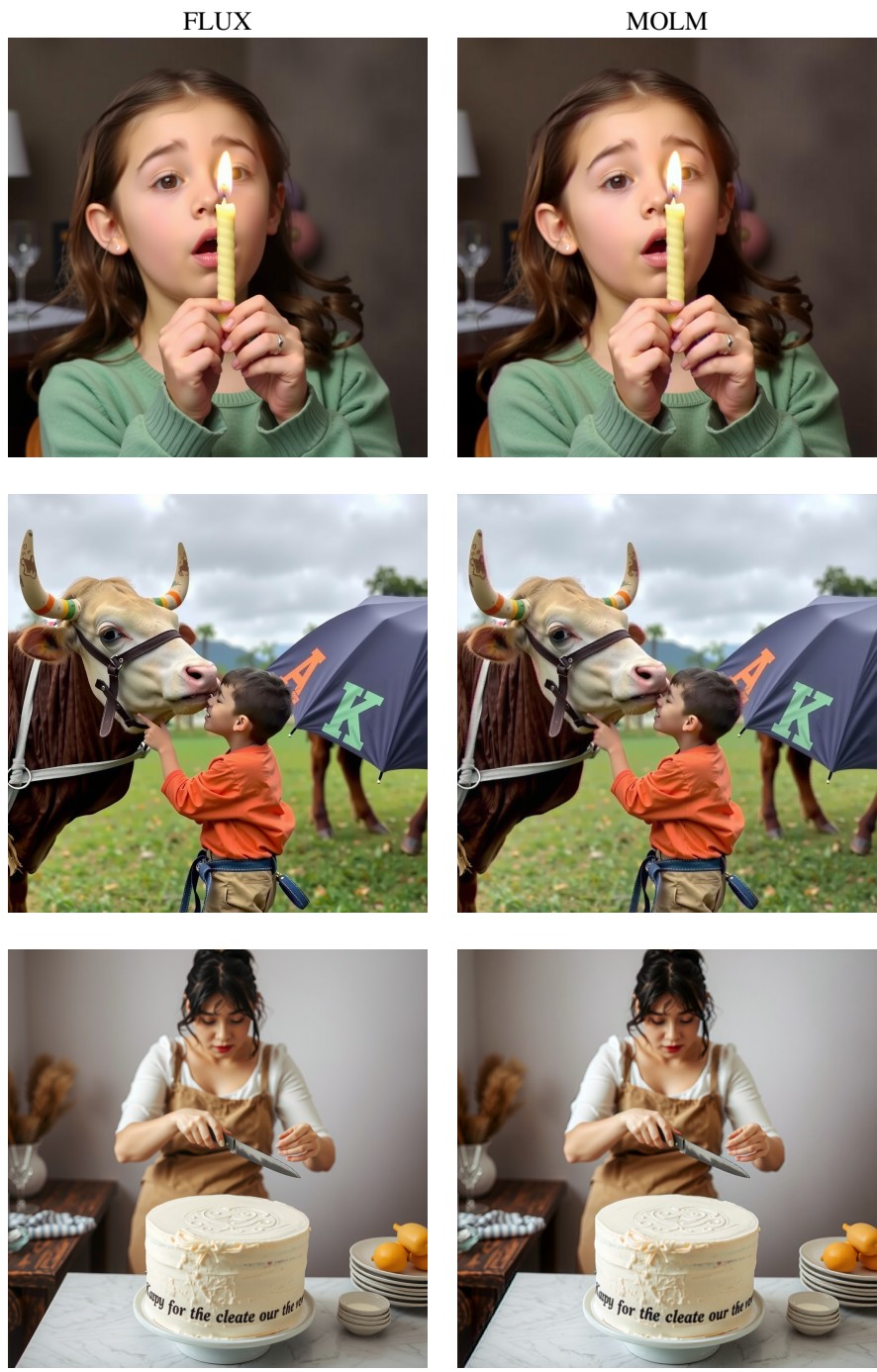

Figure 8: **Qualitative comparison on FLUX.** Side-by-side generations from FLUX (left) and MOLM (right) on MS COCO prompts. MOLM preserves the visual fidelity of FLUX while embedding a recoverable watermark.

SD 3.5          MOLM

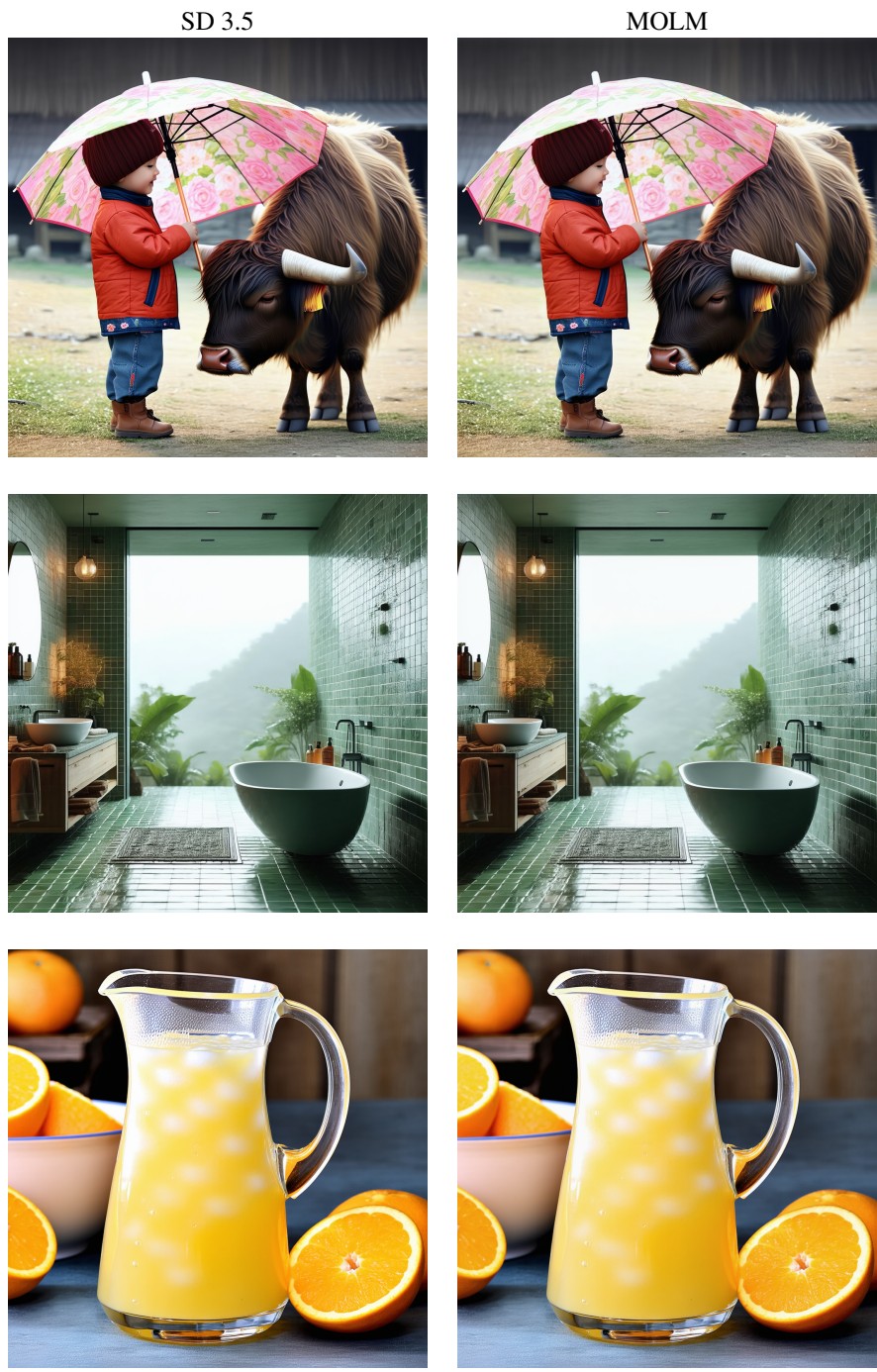

Figure 9: **Qualitative comparison on SD 3.5.** Side-by-side generations from SD 3.5 (left) and MOLM (right) on MS-COCO prompts. MOLM preserves the visual fidelity of SD 3.5 while embedding a recoverable watermark.

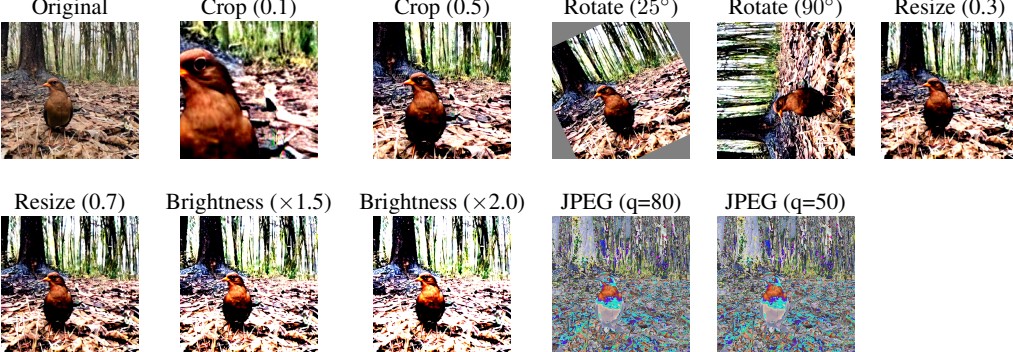

Figure 10: **Effect of distortions.** MOLM generations from MS COCO under a variety of distortions applied during evaluation. These include geometric transformations such as random cropping ($10\%$ and $50\%$) and rotations ($25°$, $90°$), resizing to different scales ($0.3\times$, $0.7\times$), photometric changes such as brightness adjustment ($\times1.5$, $\times2.0$), and lossy compression using JPEG (quality 80 and 50).

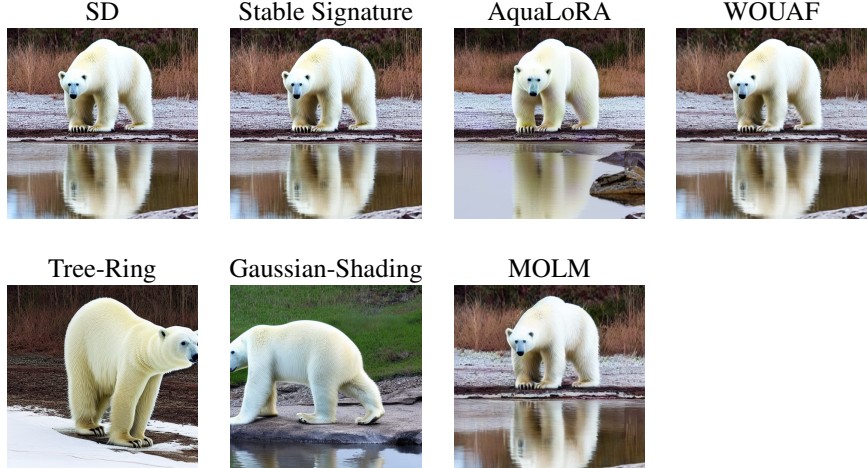

Figure 11: **Comparison with existing methods.** Visual examples from MS COCO comparing baseline Stable Diffusion (no watermark), Stable Signature, AquaLoRA, WOUAF, Tree-Ring, Gaussian-Shading, and our proposed MOLM. MOLM maintains fidelity while embedding a robust and recoverable watermark.

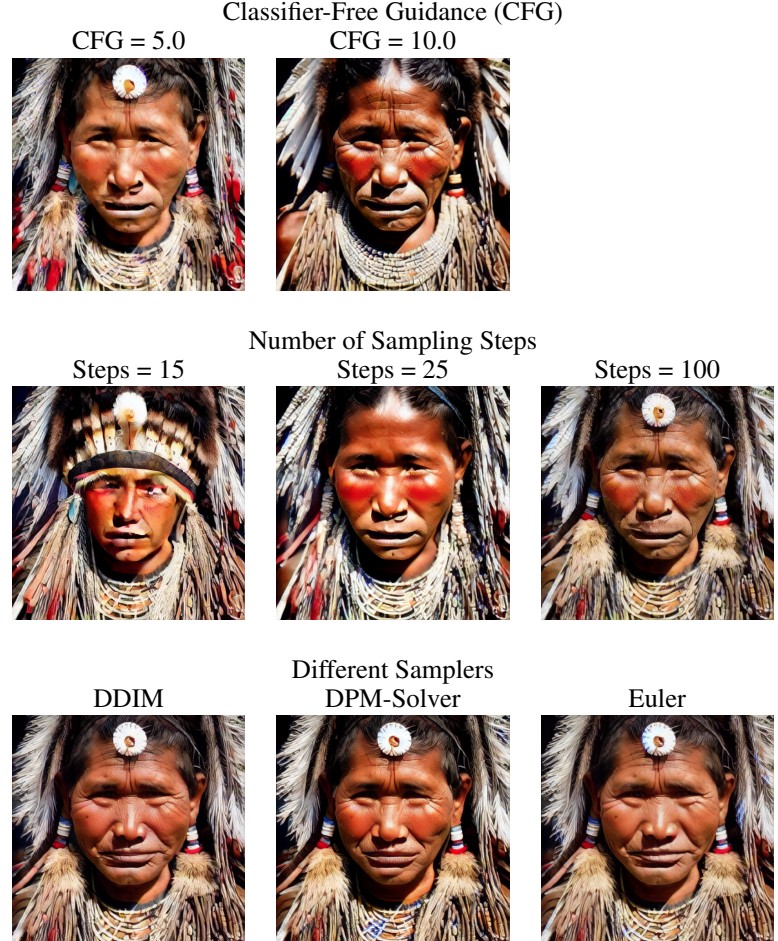

Figure 12: **Sampling configurations.** Qualitative examples from MS COCO showing MOLM generations under the sampling settings reported in Table 7: CFG scale (top), number of steps (middle), and sampler type (bottom). Across all settings, watermarks remain consistently recoverable while maintaining fidelity.

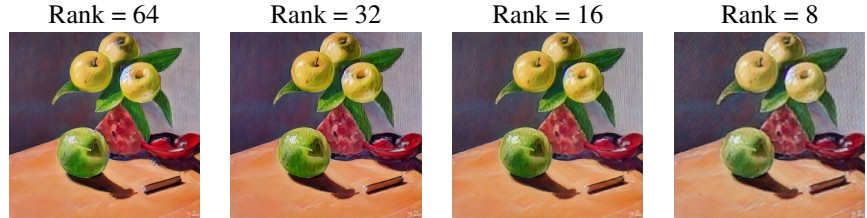

Figure 13: **LoRA rank ablation.** Qualitative examples from MS COCO showing MOLM generations with different LoRA adapter ranks. Higher ranks preserve watermark recovery more reliably.

