# OpenReview forum: "MOLM: Mixture of LoRA Markers"
_ICLR.cc/2026/Conference — ICLR 2026 Poster_

### Official Review · Reviewer_D7NA · 2025-10-27

**Soundness:** 2
**Presentation:** 3
**Contribution:** 2
**Rating:** 4
**Confidence:** 3

**Summary:**

MOLM repurposes multiple LoRA adapters as watermark “markers” and uses key-dependent routing (MoLE-style) to embed a bitstring while keeping the backbone frozen. A learned extractor recovers bits. Experiments on SD-1.5 and FLUX report near-zero inference overhead, small FID changes, and robustness to common distortions and selected white-box attacks.

**Strengths:**

1.Practicality. Lightweight LoRA adapters, no architecture changes, negligible runtime overhead.
2.Coverage. Evaluation on SD-1.5/FLUX with ~28-bit default capacity and higher-capacity variants; ablations on placement/configuration.
3.obustness. High bit accuracy under JPEG/crop/resize and diffusion regeneration; PGD-style white-box results show resilience.

**Weaknesses:**

1. The method is a straightforward combination of existing techniques (LoRA + MoE routing).
2. Fails to compare with recent state-of-the-art watermarking methods.

**Questions:**

1.Novelty. The core idea of MOLM is to repurpose LoRA adapters as watermark carriers and use key-driven routing to activate specific adapters for embedding. This is essentially a composition of mature techniques: 1) LoRA is a well-established parameter-efficient fine-tuning method; MOLM merely reinterprets it as a watermark carrier. 2) The routing logic follows MoLE, replacing data-gated expert selection with key-gated selection.
2.Baselines. Include recent watermarks or clearly justify exclusion.
3.Presentation. Fix typos (e.g., “Deployment”; “~8 days”), align the Table-3 title with its contents ('inference' is not listed, although it is mentioned in the paper), and unify notation/spacing across equations and tables.

---

> ### Author Response · Authors · 2025-11-22
>
> We thank the reviewer for taking the time to review our work and for the suggestions given to improve the paper further. We address each of the concerns below.
>
> > **W1 & Q1**: The method is a straightforward combination of existing techniques (LoRA + MoE routing). The core idea of MOLM is to repurpose LoRA adapters as watermark carriers and use key-driven routing to activate specific adapters for embedding. This is essentially a composition of mature techniques: 1) LoRA is a well-established parameter-efficient fine-tuning method; MOLM merely reinterprets it as a watermark carrier. 2) The routing logic follows MoLE, replacing data-gated expert selection with key-gated selection.
>
> We respectfully disagree with the reviewer on the lack of novelty. As clearly stated in the abstract and introduction, the primary contribution of this work is a novel watermarking framework based on key-dependent perturbations of a frozen generative model. While this can be implemented even via full fine-tuning of the generator (and we have tried this first), such an approach involves a high training cost and high storage cost (to store all the perturbed parameters). The Mixture of LoRA Experts (MoLE) idea enables us to achieve the same effect in a parameter-efficient manner. Furthermore, we would like to emphasize that the proposed method is not a straightforward adaptation of existing techniques (LoRA + MoE) for watermarking. We just use the idea of replacing the full matrix experts with a low-rank structure to save in terms of compute and storage, which has not been attempted thus far in any context. Thus, LoRA + MoE idea is just a tool to achieve parameter efficiency during watermark encoding.
>
> > **W2 & Q2**: Fails to compare with recent state-of-the-art watermarking methods. Baselines. Include recent watermarks or clearly justify exclusion.
>
> Our benchmarking already covers recent watermarking methods spanning 2023–2025: Stable Signature (2023), Tree-Ring (2023), Gaussian Shading (2024), AquaLoRA (2024), WOUAF (2024), and ROBIN (2025). Works without released implementations or targeting a different threat model are cited but excluded from the table. If there are specific recent methods the reviewer has in mind, we would be happy to include them.
>
> > **Q3**: Presentation. Fix typos (e.g., “Deployment”; “~8 days”), align the Table-3 title with its contents ('inference' is not listed, although it is mentioned in the paper), and unify notation/spacing across equations and tables.
>
> We apologize for these presentation issues and have fixed them in the revised paper.

---

> > ### Author Response · Authors · 2025-11-27
> >
> > Thank you again for your thoughtful feedback. As the discussion period is ending soon, we would greatly appreciate hearing whether our rebuttal addressed your concerns, and we are happy to respond promptly to any additional comments.

---

### Official Review · Reviewer_Tahr · 2025-10-27

**Soundness:** 2
**Presentation:** 2
**Contribution:** 2
**Rating:** 4
**Confidence:** 3

**Summary:**

The paper proposes MOLM, a watermarking framework for text-to-image diffusion models that views watermarking as key-dependent parameter perturbations of a *frozen* generator. Concretely, the method installs LoRA adapters in selected residual/attention blocks and routes activation through one adapter per block according to chunks of a binary key. Thus, a key defines a deterministic execution path (the perturbation), while the backbone weights remain unchanged. A lightweight extractor network recovers the key from generated images. The training objective balances a perceptual loss for imperceptibility with a bitwise BCE for verifiability. Experiments on Stable Diffusion v1.5 (512×512) and FLUX (1024×1024) show high key recovery, small FID deltas (≤ ~1.5), and robustness to a range of common image distortions, learned compression, diffusion-based regeneration, averaging, and white-box PGD attacks. Default configuration routesL=14 decoder ResNet blocks with P=4 adapters per block, yielding M=28-bit keys; larger capacities are discussed via more blocks or choices

**Strengths:**

1. Casting watermarking as key-conditioned parameter routing over a frozen backbone is elegant, modular, and orthogonal to model architectures. The idea neatly ties together LoRA efficiency with watermarking needs.
2. Keys correspond to routing masks; adapting capacity does not require retraining the backbone. The paper reports ~1 GPU-day one-time training and no additional inference cost beyond activating chosen LoRA paths.

**Weaknesses:**

1. The paper frames both detection and attribution but largely evaluates bit recovery / TPR against a fixed key/threshold. How MOLM behaves with large key databases (collisions, nearest-neighbor attribution errors, false-match rate under heavy post-processing) is not fully quantified.
2. The white-box attacks target the extractor in image space. Stronger adversaries could attempt prompt-, noise-, or sampler-level optimization to suppress/flip bits while staying on-distribution (e.g., plug-and-play adversaries).
3. While MOLM avoids per-key retraining, operational details (e.g., revoking/rotating keys, multi-tenant key assignment, rate-limited leakage scenarios, collusion of users averaging many outputs with heterogeneous keys) are not empirically explored.
4. The table compares to Stable Signature, AQuaLoRA, WOUAF (bit recovery) and Tree-Ring/ROBIN/Gaussian Shading (detection). Some strong recent defenses/attacks (esp. post-2024 SoK/benchmarks) may warrant a tighter apples-to-apples setup (identical prompts/seeds, extractors retrained with the same aug sets), though the paper cites them.
5. The robust baselines for watermarking arbitrary images are lacking, such as TrustMark [1], VINE [2], StegaStamp [3]. Incorporating those methods would make the comparison more thorough and robust.


[1] TrustMark: Universal Watermarking for Arbitrary Resolution Images

[2] Robust Watermarking Using Generative Priors Against Image Editing: From Benchmarking to Advances

[3] StegaStamp: Invisible Hyperlinks in Physical Photographs

**Questions:**

1. If a key leaks, how quickly can you rotate without retraining? Is there support for *soft revocation* (e.g., attenuating adapters) versus installing fresh adapter banks?
2. You evaluate averaging removal/forgery with the same messages; what about heterogeneous-key collusion across many users? Can mixed-key averaging partially cancel the watermark?
3. Have you tried prompt/noise optimization targeting the extractor gradient via a differentiable proxy, or classifier-free guidance tuning to hide bits while maintaining content?

---

> ### Author Response · Authors · 2025-11-22
>
> We thank the reviewer for taking the time to review our paper and for highlighting that our approach to watermarking is elegant, modular, and works across model architectures. Below, we address each of your concerns.
>
> > **W1**: The paper frames both detection and attribution but largely evaluates bit recovery / TPR against a fixed key/threshold. How MOLM behaves with large key databases (collisions, nearest-neighbor attribution errors, false-match rate under heavy post-processing) is not fully quantified.
>
> We thank the reviewer for this important suggestion. We have added a comprehensive attribution evaluation in the revised manuscript.
>
> **New content added:** Section "Attribution Results" (page 8) with Table 2 presents an attribution study addressing the reviewer's concern.
>
> We simulate a multi-user deployment with N = 1,000 registered users, each assigned a unique M = 28-bit key. For each user,  we generate 10 watermarked images, yielding a test set of 10,000 in-database samples. We calibrate the acceptance threshold τ = 27 (at most one bit error) to achieve a global false positive rate α = 10^-3.
>
> **Key findings:**
> - Attribution accuracy: 98.92% on clean watermarked images
> - False negative rate: 1.08% (legitimate images incorrectly rejected)
> - False positive rate: 0.02% when tested on 20,000 non-watermarked images
> - Among accepted images, attribution accuracy is ~100%
>
> **Robustness under distortions:** We evaluate attribution under the same post-processing suite used in detection experiments. Common image edits preserve attribution: JPEG compression (quality 80), mild cropping (5%), resizing (0.7×), moderate rotation (25°), and brightness adjustment (×1.5) all achieve >85% acceptance with near-perfect accuracy among accepted images. Severe transformations (0.3× resize, 90° rotation) are appropriately rejected (<1% acceptance).
>
> **Threshold trade-offs:** Our choice of τ = 27 is intentionally conservative, prioritizing low false positives for high-stakes applications. In scenarios where false negatives are more costly, τ can be relaxed to 26 or lower without retraining, demonstrating MOLM's operational flexibility.
>
> These results demonstrate that MOLM scales effectively to multi-user deployments, maintaining robust attribution under realistic editing scenarios while reliably rejecting non-watermarked content.
>
> **Collision analysis** (key design): We can assign user keys from a minimum-distance codebook 𝐶⊂{0, 1} 𝑀 with pairwise Hamming distance at least 𝑑_𝑚𝑖𝑛. This explicit margin between keys suppresses nearest-neighbor ambiguities and makes accidental collisions unlikely under the binomial bit-error model, while also stabilizing the reject option for out-of-database queries.
>
> Operationally, we can sample candidate keys uniformly and admit them only if they satisfy the minimum Hamming distance constraint (otherwise we can resample/rotate). The trade-off is a lower code rate (i.e., a reduced usable key space).
>
>
>
> > **W2**: The white-box attacks target the extractor in image space. Stronger adversaries could attempt prompt-, noise-, or sampler-level optimization to suppress/flip bits while staying on-distribution (e.g., plug-and-play adversaries).
>
> > **Q3**: Have you tried prompt/noise optimization targeting the extractor gradient via a differentiable proxy, or classifier-free guidance tuning to hide bits while maintaining content?
>
> We thank the reviewer for the suggestion. As discussed in the Problem Setting (beginning of Section 3), the threat model assumes that the adversary does not have access to any of the model parameters (complete black-box setting) and only observes images.
>
> Our evaluation in Section 4.6.3 already goes one step beyond the stated threat model – the adversary may know the extractor (verifier is compromised), but has no access to the generator’s weights, activations, gradients, or internal noise state. Accordingly, the adversarial attack experiments in Section 4.6.3 are white-box w.r.t. the extractor and optimize in the image space.
>
> Prompt-/noise-/sampler-level optimization requires gradients or privileged control over the generator through the denoiser and scheduler. This means a different threat model where either the model-owner is also compromised or there is a black-box attack on the generator via a suitable surrogate generative model. We have not considered such attacks in this paper, but will explore them in the future.

---

> > ### Comment · Reviewer_Tahr · 2025-11-23
> >
> > Thank you very much for your detailed and patient response. The authors have added many experiments and provided a strong rebuttal. I still have several concerns, outlined below:
> >
> > - Regarding Q3, I remain unconvinced. It seems that an attacker could readily obtain a suitable surrogate generative model. Could you elaborate further on why such attacks are not considered?
> >
> > - The key size of TrustMark reported in Table 1 appears to be incorrect; it should be 100. In addition, TrustMark is not the most robust among the three methods.
> >
> > - I am also not fully convinced by the advantages offered by the proposed framework. I am curious whether the framework remains robust against diffusion-based watermark removal methods, such as image regeneration or text-driven image editing (e.g., InstructPix2Pix).

---

> ### Author Response · Authors · 2025-11-22
>
> > **W3**: While MOLM avoids per-key retraining, operational details (e.g., revoking/rotating keys, multi-tenant key assignment, rate-limited leakage scenarios, collusion of users averaging many outputs with heterogeneous keys) are not empirically explored.
>
> > **Q1**: If a key leaks, how quickly can you rotate without retraining? Is there support for soft revocation (e.g., attenuating adapters) versus installing fresh adapter banks?
>
> - Key rotation and revocation do not require any retraining. MOLM adapters are shared across users. If a key leaks, we can rotate by issuing a new bitstring and updating the registry; the leaked key can be moved to a denylist and can be rejected by the verifier. This is a registry-level change, so no generator or adapter retraining is required, and there is no change to the inference pipeline. Because adapters are shared, per-key “attenuation” is not meaningful - we rely on hard revocation via the registry.
>
> The limitation of the above approach is that the new key applies only to freshly generated images after key revocation. It is not practical to recall all images that were generated, watermarked, and distributed in the past and embed them with the new key. Hence, it may no longer be possible to authenticate such past images.
>
> - Multi-tenant assignment: We allocate keys from disjoint codebook partitions and enforce a minimum Hamming distance 𝑑_𝑚𝑖𝑛 within each partition. This ensures cross-tenant disjointness and reduces nearest-neighbor ambiguities.
>
> > **Q2**: You evaluate averaging removal/forgery with the same messages; what about heterogeneous-key collusion across many users? Can mixed-key averaging partially cancel the watermark?
>
> The revised pdf includes Figure 5 (Appendix C.1), which reports MOLM bit accuracy under removal averaging attacks in the heterogeneous-key setting. Note that the standard forgery attack is not applicable here, since each image carries its own key. As shown, MOLM maintains high bit accuracy even when averaging up to 5,000 images.
>
> > **W4**: The table compares to Stable Signature, AQuaLoRA, WOUAF (bit recovery) and Tree-Ring/ROBIN/Gaussian Shading (detection). Some strong recent defenses/attacks (esp. post-2024 SoK/benchmarks) may warrant a tighter apples-to-apples setup (identical prompts/seeds, extractors retrained with the same aug sets), though the paper cites them.
>
> All methods have been evaluated under the same prompts, seeds, number of steps, and resolution. Unless a baseline mandates a specific sampler, we use the same sampler across methods; for baselines that rely on DDIM inversion, we keep the DDIM sampler as required by their detection method. When a baseline includes a trainable extractor, we follow the authors’ published code: e.g., WOUAF is retrained with its published aug set; AquaLoRA is evaluated using the released models. Tree-Ring and Gaussian Shading require no training. ROBIN is run with per-prompt adversarial optimization using the authors’ public code. In inference, we apply an identical distortion suite to all methods. Each method is verified using its own detection rules.
>
>
> > **W5**: The robust baselines for watermarking arbitrary images are lacking, such as TrustMark [1], VINE [2], StegaStamp [3]. Incorporating those methods would make the comparison more thorough and robust.
>
> Thank you for the suggestion. Our work targets in-generation watermarking for text-to-image diffusion. Methods such as TrustMark, VINE, and StegaStamp are post-hoc watermarking schemes designed for arbitrary images (and, in StegaStamp’s case, the print–capture channel), rather than watermarking during diffusion sampling or after generation. As such, their threat models and goals differ from this paper.
>
> Despite this difference, to strengthen the comparison, we evaluated TrustMark using their released model by applying it post hoc to clean SD-1.5 generations. We then measure the TPR, as TrustMark only provides a valid/invalid binary decision. These results are included in Table 1 (new rows highlighted in magenta).

---

> ### Author Response · Authors · 2025-11-25
>
> > Q1: Regarding Q3, I remain unconvinced. It seems that an attacker could readily obtain a suitable surrogate generative model. Could you elaborate further on why such attacks are not considered?
>
> It is true that the attacker can readily obtain a suitable surrogate generative model. Suppose that the attacker also has white-box access to the watermark extractor and a surrogate generator. Under these strong assumptions, it is not clear what is the best possible attack strategy. As suggested by the reviewer, we have tried optimizing the prompt embedding or the initial (latent) noise of the surrogate generator. However, these simple strategies fail to produce any advantage over the white-box attack on the watermark extractor carried out in the image (pixel) space (without any access to the generator).
> At the same time, we do not wish to overclaim that our proposed method is robust under this strong threat model because it may be possible to design a better attack strategy. For example, given (i) sufficient number of non-watermarked images from the surrogate generator, (ii) corresponding (generated using the same prompt) watermarked images from the target generator, and (iii) white-box access to the watermark extractor, can we use the same MOLM approach to learn key-dependent perturbations of the frozen surrogate generator that mimics the MOLM (watermark encoder) of the target generator? What is the transferability of such an approach? These are deeper questions that require further investigation and will be considered in future work.
>
> We would like to emphasize again that the security of any system should be analyzed based on a well-defined threat model. We have already demonstrated that the proposed approach is robust against the defined threat model (no white-box access to generator or watermark extractor parameters). We went one step further and established that the approach is also robust even when the attacker has white-box access to the extractor. At this point, we do not wish to make any other stronger claims.
>
> > Q2: The key size of TrustMark reported in Table 1 appears to be incorrect; it should be 100. In addition, TrustMark is not the most robust among the three methods.
>
> We apologize for the oversight. TrustMark indeed embeds a fixed 100-bit codeword. The string "mysecret" came directly from the project’s demo script;  under the default BCH\_5 ECC, this fits within the user-payload budget, but the raw payload remains 100 bits.
> We also ran VINE on the same image set we used for TrustMark. For VINE, mean bit-accuracy for each attack was: clean 1.000, jpeg 1.000, brightness 0.951, rot 0.514, crop 0.519. Vine is more robust than TrustMark on our suite, but it is still vulnerable to cropping and rotation. We will add these results to Table 1.
>
> > Q3: I am also not fully convinced by the advantages offered by the proposed framework. I am curious whether the framework remains robust against diffusion-based watermark removal methods, such as image regeneration or text-driven image editing (e.g., InstructPix2Pix).
>
> We would like to clarify that diffusion-based watermark removal attacks **are already evaluated in Section 4.7.1** of the original submission. **Table 3 (Row 3)** of the original submission presents results for diffusion-based regeneration attacks. In this attack, an adversary attempts to remove the watermark by passing the image through DDIM inversion and denoising reconstruction.
> In response to the reviewer's specific suggestion, we have now evaluated MOLM against InstructPix2Pix, a text-driven image editing model. We define three severity levels of prompts based on the degree of semantic modification: Mild: Color/lighting adjustments, Moderate: Style/texture changes, and Severe: Structural/semantic changes. For each of the 100 watermarked images, we apply two prompts per severity level using InstructPix2Pix with standard hyperparameters (image guidance scale = 1.5, text guidance scale = 7.5, 50 inference steps). Bit accuracy remains robust at 0.81.

---

> > ### Comment · Reviewer_Tahr · 2025-11-26
> >
> > Thank you for the detailed clarifications. You have addressed all three of my concerns. the discussion regarding surrogate generator (Q1), the corrected TrustMark key size and added more results to support (Q2), and the existing plus newly added evaluations on diffusion-based removal methods including InstructPix2Pix (Q3).
> >
> > These responses resolve my concerns. I'm currently inclined to raise my score, but I'll wait and see what is the opinion of the other reviewers. I'll give the final score at the end of the rebuttal.

---

> > > ### Author Response · Authors · 2025-11-26
> > >
> > > Thank you for the engagement and for giving us the chance to clarify our work.
> > > We appreciate your careful reading and constructive feedback, and we're glad our responses addressed your concerns.

---

### Official Review · Reviewer_yLkc · 2025-11-02

**Soundness:** 2
**Presentation:** 2
**Contribution:** 2
**Rating:** 4
**Confidence:** 3

**Summary:**

This paper addresses verifying and attributing AI-generated images. The authors propose a watermarking framework inspired by the Mixture of LoRA Experts. They formulate the problem as key-dependent perturbations of frozen generative models, where binary keys are used to route among a mixture of LoRA adapters inserted into the generative model. In addition, the authors consider imperceptibility, fidelity, verifiability, and robustness by incorporating the L_imp​ and L_ver​ losses.

**Strengths:**

The underlying ideas are reasonable.

The evaluation demonstrates imperceptibility, fidelity, verifiability, and robustness.

The paper includes ablation studies supported by experiments and visualization.

The experimental details are clearly presented.

**Weaknesses:**

The watermarking results do not clearly demonstrate state-of-the-art performance; they are only competitive with previous methods.

The description of the proposed method is generally ok, but some paragraphs are poorly written and fragmented, which affects readability.

**Questions:**

The authors mention that the proposed method is inspired by Mixture of LoRA Experts (MoLE), but the motivation behind this choice is unclear. It would be helpful to explain the intuition for why MoLE's structure is beneficial for the watermarking problem, beyond the fact that it can be adapted to it.

Diffusion version 3.5 Large (8B) has already been released on Hugging Face. However, this paper still evaluates on version 1.5. It would be more convincing to include results from a more recent version to demonstrate scalability and relevance.

In Table 1, the bolded values appear to indicate the best and second-best results within each column, but this should be explicitly explained in the table caption. The meaning of the orange-highlighted values should also be clarified.

It would strengthen the paper if more baseline methods were reimplemented and evaluated on the FLUX architecture and LAION dataset to provide a fair and comprehensive comparison with the proposed approach.

---

> ### Author Response · Authors · 2025-11-22
>
> We thank the reviewer for taking the time to review our work and for the suggestions given to improve the paper further. We address each of the concerns below.
>
> > **W1**: The watermarking results do not clearly demonstrate state-of-the-art performance; they are only competitive with previous methods.
>
> We agree with the reviewer that the proposed method is comparable to prior methods in terms of the detection and image quality metrics. However, as clearly noted in the introduction (lines 79-84), the proposed method has other advantages such as **efficiency**, **scalability**, and **robustness against strong removal and forgery attacks**. Thus, the primary contribution of this work is a new watermarking framework based on key-dependent perturbations of a frozen generative model, which has the above three advantages. We have been careful not to claim state-of-the-art performance in terms of the detection and image quality metrics.
>
> > **W2**: The description of the proposed method is generally ok, but some paragraphs are poorly written and fragmented, which affects readability.
>
> We apologize for the oversight, and we will revise the paper to improve readability.
>
> > **Q1**: The authors mention that the proposed method is inspired by Mixture of LoRA Experts (MoLE), but the motivation behind this choice is unclear. It would be helpful to explain the intuition for why MoLE's structure is beneficial for the watermarking problem, beyond the fact that it can be adapted to it.
>
> We thank the reviewer for bringing this to our attention. Our primary contribution is a watermarking framework that treats embedding as key-dependent parameter perturbations of a frozen generator. While this can be realized via full fine-tuning (we tried this first), it is costly in training and storage (to store all the perturbed parameters). We therefore implement the same idea with **low-rank adapters**, i.e., a MoLE-style layout used purely for parameter efficiency by replacing dense experts with low-rank adapters. The method is **not** a straightforward adaptation of MoLE for watermarking: the routing is fixed by the key and optimized for verifiability/imperceptibility, rather than data-gated for task performance. We clarify this rationale in the revision.
>
>
> > **Q2**: Diffusion version 3.5 Large (8B) has already been released on Hugging Face. However, this paper still evaluates on version 1.5. It would be more convincing to include results from a more recent version to demonstrate scalability and relevance.
>
> Thank you for the suggestion. We have trained MOLM on Stable Diffusion 3.5–Large (8B) using the same recipe as SD-1.5 (same routing placement and capacity, i.e., L=14 routed blocks with P =4 adapters ⇒ M = 28 bits; and same losses). The new results are added to Table 1 in the revised PDF (rows highlighted in magenta), and visual examples are provided in the Appendix (Fig. 9). We observe the same qualitative behavior as on SD-1.5: high key recovery, small fidelity changes, and comparable robustness trends across common distortions, with no added inference overhead. This further supports our claim that MOLM scales to recent T2I architectures without changing the sampler or retraining per key.
>
> > **Q3**: In Table 1, the bolded values appear to indicate the best and second-best results within each column, but this should be explicitly explained in the table caption. The meaning of the orange-highlighted values should also be clarified.
>
> We apologize for the lack of clarity. We have updated the caption for Table 1 in the revised PDF to indicate that within each column, bold marks the best value and the second-best. Orange-highlighted values report the change in fidelity relative to the non-watermarked images.
>
> > **Q4**: It would strengthen the paper if more baseline methods were reimplemented and evaluated on the FLUX architecture and LAION dataset to provide a fair and comprehensive comparison with the proposed approach.
>
> Thank you for the suggestion. We have implemented and evaluated additional methods on FLUX architecture and LAION-Aesthetics dataset under a matched protocol (identical prompts, CFG, seeds, and resolution; retrained extractors where applicable). The new results are included in Table 1 of the revised PDF (rows highlighted in magenta).  It must be noted that FLUX cannot be implemented in the following baseline methods: Tree-Ring uses an invertible stable diffusion pipeline (U-Net pipelines). FLUX uses a different architecture (transformer-based) and different sampling/inversion mechanics, so to run Tree-Ring on FLUX, we need an inversion method for rectified flow and adapt the ring embedding/detection to FLUX latents/schedule. For the same reasons, Gaussian-Shading and ROBIN cannot be used with FLUX. AquaLoRa encodes a watermark via the U-Net, and FLUX uses a rectified-flow transformer with a different module layout.

---

> > ### Author Response · Authors · 2025-11-27
> >
> > Thank you again for your thoughtful feedback. As the discussion period is ending soon, we would greatly appreciate hearing whether our rebuttal addressed your concerns, and we are happy to respond promptly to any additional comments.

---

### Author Response · Authors · 2025-11-22
**Global Response**

We thank all reviewers for their thoughtful feedback. We are encouraged that they found the core idea, casting watermarking as key-conditioned routing over a frozen backbone, both simple and broadly applicable.  We also appreciate the recognition of the method’s practicality (lightweight adapters, no architectural changes), the breadth of our evaluation (imperceptibility, fidelity, verifiability, robustness). The robustness under common post-processing and diffusion regeneration, as well as resilience to white-box attacks, was noted positively. Finally, we are pleased that reviewers found the experimental details to be clear and reproducible.

What we addressed in the revised pdf/rebuttal:
- On design rational and novelty: We are not “combining LoRA with MoLE.” Our framework is watermarking as key‑dependent parameter perturbations, $U_{\zeta}(G_{\Phi}(q,t),\kappa)=G_{\Phi+\Delta\Phi(\kappa)}(q,t)$. We first realized $\Delta\Phi(\kappa)$ via full fine‑tuning with multiple experts, but this was costly; LoRA is used purely to make it practical, replacing $m\times n$ updates with $(m+n)\,r$ factors (small $r$). The binary key $\kappa$ deterministically selects one adapter per routed block.
- Attribution at scale: We add Attribution (multi‑user) to the metrics and a new Attribution Results section. We obtain: Attribution Accuracy **98.92\%**. Common edits largely preserve attribution; severe transforms are rejected.
- Modern backbones. We add SD\,3.5‑Large (8B) results and visuals (Table~1 and Appendix), complementing SD‑1.5/FLUX and showing similar behavior (high recovery and quality).
- On baselines. We expand baselines on FLUX, if applicable, and LAION-Aesthetics. For post‑hoc arbitrary image watermarking methods, we report TrustMark on our generated images.

We are grateful for the reviews. The revisions clarify the motivation, strengthen baselines on modern models/datasets, tighten presentation, and add an attribution‑at‑scale experiments, while preserving the practicality and modularity noted by the reviewers.

---

### Comment · Area_Chair_XGmk · 2025-11-25
**Authors' responses**

Dear Reviewers,

The authors have submitted their responses to your questions and feedbacks. Please read them and give your comments.

Regards,
AC

---

### Comment · Area_Chair_XGmk · 2025-11-28
**The Author/Reviewer Discussion Phase deadline is approaching.**

Dear Reviewers,

The Author/Reviewer Discussion Phase deadline is approaching. If you have not responded to authors’ rebuttal, please read and give your feedback asap.

Regards,
AC

---

### Author Response · Authors · 2025-12-03
**Rebuttal Summary**

We thank the reviewers for their time and care devoted to evaluating our work. Below, we give a concise summary of the main points addressed in the rebuttal. We replied to all reviews; **Reviewer Tahr** kindly participated in the discussion and wrote that our answers resolved their concerns and that they are **inclined to raise their score**, while the other two reviewers did not follow up.

---

## **Overall strengths**

- **Reviewer yLkc** values that the paper’s ideas are reasonable and that the experiments jointly study imperceptibility, fidelity, verifiability, and robustness, supported by clear ablations and implementation details.
- **Reviewer Tahr** emphasizes the elegance and modularity of key-conditioned routing over a frozen generator, and appreciates that the approach is efficient in practice (no per-key retraining, no extra inference cost).
- **Reviewer D7NA** highlights the practicality of using lightweight LoRA adapters with no architectural changes, the coverage across SD-1.5 and FLUX with multiple capacities, and the observed robustness to common image perturbations and selected attacks.

Together, these comments recognize a practical and broadly evaluated framework for watermarking generative models via key-dependent parameter perturbations.

---

## **Main points from the rebuttal**

### **Reviewer yLkc**

- **[W1] “not clearly SOTA”.**
  We clarified that our main contribution is a **watermarking framework based on key-dependent parameter perturbations** of a frozen generator; our goal is not to claim metric SOTA but to provide an efficient, scalable, and robust design.

- **[W2] Writing and organization.**
  We edited the manuscript to improve readability, smoothing fragmented paragraphs and clarifying definitions and notation.

- **[W3] Use of modern backbones.**
  We added experiments on **SD-3.5-Large**, showing that the proposed routing scheme and capacity choices transfer to a more recent, larger backbone without additional inference overhead.

- **[W4] Baselines on FLUX / LAION.**
  We expanded baselines wherever they are technically compatible with FLUX/LAION and explained why some U-Net/DDIM-specific methods cannot be fairly ported, in order to keep comparisons consistent.

---

### **Reviewer Tahr**

- **[W1] Attribution at scale.**
  We introduced a **large-scale attribution study** (many users and images) with a conservative threshold, reporting high attribution accuracy and low false-positive rates, including under typical image edits.

- **[W2] Strength of adversaries and threat model.**
  We made the **no-generator-access threat model** explicit, evaluated strong image-space white-box attacks against the extractor, and reported surrogate-based attempts, avoiding claims beyond this setting.

- **[W3] Operational questions (key rotation, multi-tenant use, collusion).**
  We described how keys can be **rotated or revoked** at the registry level without retraining, how **disjoint codebooks** enable multi-tenant deployments, and how heterogeneous-key averaging behaves when combining many images.

- **[W4] Fairness of baseline comparisons.**
  We detailed that all methods use **identical prompts, seeds, step counts, and resolution**; we respect each baseline’s required sampler/training and apply a common distortion suite while using each method’s own detection rule.

- **[W5] Additional robust post-hoc baselines.**
  Although these methods target arbitrary existing images and a slightly different threat model, we added comparisons with TrustMark (with corrected 100-bit payload) and VINE on our generated images and discussed their strengths and remaining vulnerabilities.

**`After this exchange, Reviewer Tahr stated that our responses resolved their concerns and that they are inclined to raise their score.`**

---

### **Reviewer D7NA**

- **[W1] Perceived novelty (“LoRA + MoE recombination”).**
  We emphasized that the central idea is **watermarking via key-dependent parameter perturbations** of a frozen generator; LoRA/MoLE is used purely as a parameter-efficient way to implement this, rather than being the contribution itself.

- **[W2] Coverage of recent baselines.**
  We highlighted and modestly extended our inclusion of recent watermarking methods where public code and compatible settings are available, and acknowledged additional methods that cannot yet be compared fairly.

- **[W3] Presentation and minor issues.**
  We corrected the reported typos, adjusted table titles, and cleaned up notation and spacing.

---

We believe these changes and clarifications substantially strengthen both the clarity and the technical scope of the work, and we are very grateful to the reviewers and committee for their feedback and consideration.

— The Authors

---

### Meta-Review · Area_Chair_NVQo · 2026-01-09

**Summary:**

The paper proposes an adaptation of Mixture-of-LORA-experts to the watermarking problem.

yLkc: Score 4: Results are not SOTA + Poor writing
Tahr: Score 4: Missing baselines, experimental validation concerns (hyperparameters, prompt, etc.)
D7NA: Score 4: 'Not SOTA' (very short review)

**Reviewer Concerns:**

Reviewers yLkc and D7NA expressed very short concerns even without references (not SOTA,... poor writing).
Reviewer Tahr has written a comprehensive review, participated in the discussion and in the rebuttal the authors added a lot of information; he/she also was inclined to raise the score, two other reviewers did not participate in the discussion. So I think all concerns have been addressed.

**Reviewer Scores:**

yLkc: 4 -> 6
Tahr: 4 -> 6
D7NA: 4 -> 6

---

### Decision · Program_Chairs · 2026-01-26

Accept (Poster)